# BRIDGING THE IMITATION GAP BY ADAPTIVE INSUBORDINATION

## ABSTRACT

When expert supervision is available, practitioners often use imitation learning with varying degrees of success. We show that when an expert has access to privileged information that is unavailable to the student, this information is marginalized in the student policy during imitation learning resulting in an "imitation gap" and, potentially, poor results. Prior work bridges this gap via a progression from imitation learning to reinforcement learning. While often successful, gradual progression fails for tasks that require frequent switches between exploration and memorization skills. To better address these tasks and alleviate the imitation gap we propose 'Adaptive Insubordination' (ADVISOR), which dynamically weights imitation and reward-based reinforcement learning losses during training, enabling switching between imitation and exploration. On a suite of challenging didactic and MINIGRID tasks, we show that ADVISOR outperforms pure imitation, pure reinforcement learning, as well as their sequential and parallel combinations.

## 1 INTRODUCTION

Imitation learning (IL) can be remarkably successful in settings where reinforcement learning (RL) struggles. For instance, IL succeeds in complex tasks with sparse rewards (Chevalier-Boisvert et al., 2018a; Peng et al., 2018; Nair et al., 2018), and when the observations are high-dimensional, *e.g.*, in visual 3D environments (Kolve et al., 2019; Savva et al., 2019). In such tasks, obtaining a high quality policy purely from reward-based RL is often challenging, requiring extensive reward shaping and careful tuning as reward variance remains high. In contrast, IL leverages an expert which is generally less impacted by the environment's random state. However, designing an expert often relies on privileged information that is unavailable at inference time. For instance, it is straightforward to create a navigational expert when privileged with access to a connectivity graph of the environment (using shortest-path algorithms) (*e.g.*, Gupta et al., 2017b) or an instruction-following expert which leverages an available semantic map (*e.g.*, Shridhar et al., 2020; Das et al., 2018b). Similarly, game experts may have the privilege of seeing rollouts (Silver et al., 2016) or vision-based driving experts may have access to ground-truth layout (Chen et al., 2020). Such graphs, maps, rollouts, or layouts aren't available to the student or at inference time.

How does use of a privileged expert influence the student policy? We show that training an agent to imitate such an expert results in a policy which marginalizes out the privileged information. This can result in a student policy which is sub-optimal, and even near-uniform, over a large collection of states. We call this discrepancy between the expert policy and the student policy the *imitation gap*. A frequent strategy used in prior work to improve upon expert demonstrations (and implicitly to overcome the imitation gap when applicable) is stage-wise training: IL is used to 'warm start' learning and subsequent reward-based RL algorithms, such as proximal policy optimization (PPO), are used for fine-tuning (Lowe et al., 2020). While this strategy is often successful, the following example shows that it can fail dramatically.

**Example 1** (Poisoned Doors). Suppose an agent is presented with $N \geq 3$ doors $d_1, \ldots, d_N$. As illustrated in Fig. 1 (for $N = 4$), opening $d_1$ requires entering an unknown but fixed code of length $M$. Successful code entry results in a guaranteed reward of 1, otherwise the reward is 0. Since the code is unknown to the agent, it would have to learn the code. All other doors can be opened without a code.

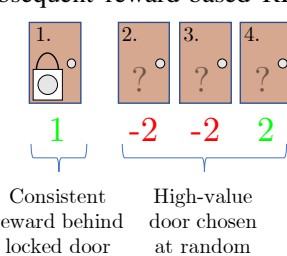

Figure 1: PoisonedDoors

For some randomly chosen $2 \leq j \leq N$ (sampled each episode), the
reward behind $d_j$ is 2 but for all $i \in \{2, \ldots, N\} \setminus j$ the reward behind $d_i$ is $-2$. Without knowledge of $j$, the optimal policy is to always enter the correct code to open $d_1$ obtaining an expected reward of 1. In contrast, if the expert is given the privileged knowledge of the door $d_j$ with reward 2, it will always choose to open this door immediately. It is easy to see that an agent without knowledge of $j$ attempting to imitate such an expert will learn open a door among $d_2, \ldots, d_N$ uniformly at random obtaining an expected return of $-2 \cdot (N-3)/(N-1)$. Training with reward-based RL after this 'warm start' is strictly worse than starting without it: the agent needs to unlearn its policy and then, by chance, stumble into entering the correct code for door $d_1$, a practical impossibility when $M$ is large.

To bridge the imitation gap, we introduce **Ad**aptive **In**subor**d**ination (ADVISOR). ADVISOR adaptively weights imitation and RL losses. Specifically, throughout training we use an auxiliary actor which judges whether the current observation is better treated using an IL or a RL loss. For this, the auxiliary actor attempts to reproduce the expert's action using the observations of the student at every step. Intuitively, the weight corresponding to the IL loss is large when the auxiliary actor can reproduce the expert's action with high confidence and is otherwise small. As we show empirically, ADVISOR combines the benefits of IL and RL while avoiding the pitfalls of either method alone. Most IL algorithms were designed with common-sense but strong assumptions which implicitly disallow a discrepancy between expert and student observations: it is when these assumptions are violated (as they often are in practice) that the imitation gap appears.

We evaluate the benefits of employing ADVISOR across ten tasks including the Poisoned Doors discussed above, a 2D gridworld, and a suite of tasks based on the MINIGRID environment (Chevalier-Boisvert et al., 2018a;b). Across all tasks, ADVISOR outperforms popular IL and RL baselines as well as combinations of these methods. We also demonstrate that ADVISOR can learn to ignore corruption in expert supervision. ADVISOR can be easily incorporated into existing RL pipelines. The code to do the same is included in the supplement and will be made publicly available.

## 2 RELATED WORK

A series of solutions (*e.g.*, Mnih et al., 2015; van Hasselt et al., 2016; Bellemare et al., 2016; Schaul et al., 2016) have made off-policy deep Q-learning methods stable for complex environments like Atari Games. Several high-performance (on-policy) policy-gradient methods for deep-RL have also been proposed (Schulman et al., 2015a; Mnih et al., 2016; Levine et al., 2016; Wang et al., 2017; Silver et al., 2016). For instance, Trust Region Policy Optimization (TRPO) (Schulman et al., 2015a) improves sample-efficiency by safely integrating larger gradient steps, but is incompatible with architectures with shared parameters between policy and value approximators. Proximal Policy Optimization (PPO) (Schulman et al., 2017) employs a clipped variant of TRPO's surrogate objective and is widely adopted in the deep RL community. We also use it as a baseline in our experiments.

As environments get more complex, navigating the search space with only deep RL and simple heuristic exploration (such as $\epsilon$-greedy) is increasingly difficult, leading to methods that imitate expert information (Subramanian et al., 2016). While several approaches exist for leveraging expert feedback, *e.g.*, Cederborg et al. (2015) consider policy shaping with human evaluations, a simple, popular, approach to imitation learning (IL) is Behaviour Cloning (BC), a supervised classification loss between the policy of the learner and expert agents (Sammut et al., 1992; Bain & Sammut, 1995). BC suffers from compounding of errors due to covariate shift, namely if the learning agent makes a single mistake at inference time then it can rapidly enter settings where it has never received relevant supervision and thus fails (Ross & Bagnell, 2010). Data Aggregation (DAgger) (Ross et al., 2011) is the go-to online sampling framework that trains a sequence of learner policies by querying the expert at states beyond those that would be reached by following only expert actions. IL is further enhanced, *e.g.*, via hierarchies (Le et al., 2018), by improving over the expert (Chang et al., 2015; Brys et al., 2015; Jing et al., 2020), bypassing any intermediate reward function inference (Ho & Ermon, 2016), and/or learning from experts that differ from the learner (Gupta et al., 2017a; Jiang, 2019; Gangwani & Peng, 2020). A sequential combination of IL and RL, *i.e.*, pre-training a model on expert data before letting the agent interact with the environment, performs remarkably well. This strategy has been applied in a wide range of applications – the game of Go (Silver et al., 2016), robotic and motor skills (Pomerleau, 1991; Kober & Peters, 2009; Peters & Schaal, 2008; Rajeswaran et al., 2018), navigation in visually realistic environments (Gupta et al., 2017b; Das et al., 2018a), and web & language based tasks (He et al., 2016; Das et al., 2017; Shi et al., 2017; Wang et al., 2018).

Recent methods mix expert demonstrations with the agent's own roll-outs instead of using a sequential combination of IL followed by RL. Chemali & Lazaric (2015) perform policy iteration from expert and on-policy demonstrations. DQfD (Hester et al., 2018) initializes the replay buffer with expert episodes and adds roll-outs of (a pretrained) agent. They weight experiences based on the previous temporal difference errors (Schaul et al., 2016) and use a supervised loss to learn from the expert. For continuous action spaces, DDPGfD (Vecerík et al., 2017) is an analogous incorporation of IL into DDPG (Lillicrap et al., 2016). POfD (Kang et al., 2018) improves performance by adding a demonstration-guided exploration term, *i.e.*, the Jensen-Shannon divergence between the expert's and the learner's policy (estimated using occupancy measures). THOR uses suboptimal experts to reshape rewards and then searches over a finite planning horizon (Sun et al., 2018). Zhu et al. (2018) show that a combination of GAIL (Ho & Ermon, 2016) and RL can be highly effective for difficult manipulation tasks.

Critically, the above methods have, implicitly or explicitly, been designed under certain assumptions (*e.g.*, the agent operates in an MDP) which imply the expert and student observe the same state. Different from the above methods, we investigate the difference of privilege between the expert policy and the learned policy. Contrary to a sequential, static, or rule-based combination of supervised loss or divergence, we train an auxiliary actor to adaptively weight IL and RL losses. To the best of our knowledge, this hasn't been studied before.

Our approach attempts to address imitation directly, assuming the information available to the learning agent is fixed. An indirect approach for reducing this gap is to enrich the information available to the agent or to improve the agent's memory of past experience. Several works have considered this direction in the context of autonomous driving (Codevilla et al., 2018; Hawke et al., 2020) and continuous control (Gangwani et al., 2019). We expect that these methods can be fruitfully combined with the method that we discuss next.

## 3 ADVISOR

We first introduce notation to define the imitation gap and illustrate how it leads to 'policy averaging.' Next, using the construct of an auxiliary policy, we propose ADVISOR to bridge this gap. Finally, we show how to estimate the auxiliary policy in practice using deep nets.

### 3.1 IMITATION GAP

We want an agent to complete task $\mathcal{T}$ in environment $\mathcal{E}$. The environment has states $s \in \mathcal{S}$ and the agent executes an action $a \in \mathcal{A}$ at every discrete timestep $t \geq 0$. For simplicity and w.l.o.g. assume both $\mathcal{A}$ and $\mathcal{S}$ are finite. For example, let $\mathcal{E}$ be a 1D-gridworld in which the agent is tasked with navigating to a location by executing actions to move left or right, as shown in Fig. 2a. Here and below we assume states $s \in \mathcal{S}$ encapsulate historical information so that $s$ includes the full trajectory of the agent up to time $t \geq 0$. The objective is to find a policy $\pi$, a mapping from states to distributions over actions, which maximizes an evaluation criterion. Often this policy search is restricted to a set of feasible policies $\Pi^{\text{feas.}}$, for instance $\Pi^{\text{feas.}}$ may be the set $\{\pi(\cdot; \theta) : \theta \in \mathbb{R}^D\}$ where $\pi(\cdot; \theta)$ is a deep neural network with $D$-dimensional parameters $\theta$. In classical (deep) RL (Mnih et al., 2015; 2016), the evaluation criterion is usually the expected $\gamma$-discounted future return.

We focus on the setting of partially-observed Markov decision processes (POMDPs) where an agent makes decisions without access to the full state information. We model this restricted access by defining a *filtration function* $f : \mathcal{S} \rightarrow \mathcal{O}_f$ and limiting the space of feasible policies to those policies $\Pi_f^{\text{feas.}}$ for which the value of $\pi(s)$ depends on $s$ only through $f(s)$, *i.e.*, so that $f(s) = f(s')$ implies $\pi(s) = \pi(s')$. We call any $\pi$ satisfying this condition an *$f$-partial policy* and the set of feasible $f$-partial policies $\Pi_f^{\text{feas.}}$. In a gridworld example, $f$ might restrict $s$ to only include information local to the agent's current position as shown in Figs. 2c, 2d. If a $f$-partial policy is optimal among all other $f$-partial policies, we say it is *$f$-optimal*. We call $o \in \mathcal{O}_f$ a *partial-observation* and for any $f$-partial policy $\pi_f$ we write $\pi_f(o)$ to mean $\pi_f(s)$ if $f(s) = o$. It is frequently the case that, during training, we have access to an expert policy which is able to successfully complete the task $\mathcal{T}$. This expert policy may have access to the whole environment state and thus may be optimal among all policies. Alternatively, the expert policy may, like the student, only make decisions given partial information (*e.g.*, a human who sees exactly the same inputs as the student). For flexibility we will

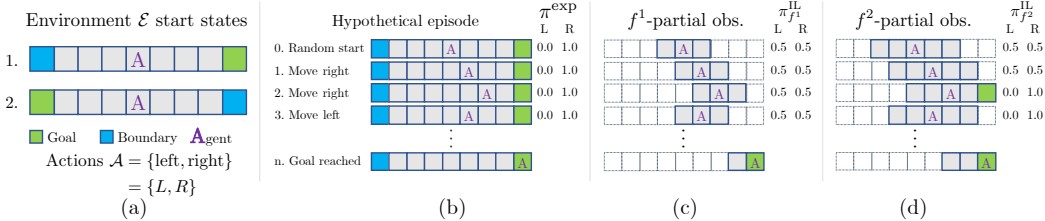

Figure 2: **Effect of partial observability in a 1-dimensional gridworld environment.** (a) The two start states and actions space for 1D-Lighthouse with $N = 4$. (b) A trajectory of the agent following a hypothetical random policy. At every trajectory step we display output probabilities as per the shortest-path expert ($\pi^{\text{exp}}$) for each state. (c/d) Using the same trajectory from (b) we highlight the partial-observations available to the agent (shaded gray) under different filtration function $f^1, f^2$. Notice that, under $f^1$, the agent does not see the goal within its first four steps. The policies $\pi_{f^1}^{\text{IL}}, \pi_{f^2}^{\text{IL}}$, learned by imitating $\pi^{\text{exp}}$, show that imitation results in sub-optimal policies *i.e.* $\pi_{f^1}^{\text{IL}}, \pi_{f^2}^{\text{IL}} \neq \pi^{\text{exp}}$.

define the expert policy as $\pi_{f^{\text{exp}}}^{\text{exp}}$, denoting it is a $f^{\text{exp}}$-partial policy for some filtration function $f^{\text{exp}}$. For simplicity, we will assume that $\pi_{f^{\text{exp}}}^{\text{exp}}$ is $f^{\text{exp}}$-optimal. Subsequently, we will drop the subscript $f^{\text{exp}}$ unless we wish to explicitly discuss multiple experts simultaneously.

In IL (Osa et al., 2018; Ross et al., 2011), $\pi_f$ is trained to mimic $\pi^{\text{exp}}$ by minimizing the (expected) cross-entropy between $\pi_f$ and $\pi^{\text{exp}}$ over a set of sampled states $s \in \mathcal{S}$:

$$\min_{\pi_f \in \Pi_f^{\text{feas.}}} \mathbb{E}_\mu[CE(\pi^{\text{exp}}, \pi_f)(S)] \,, \tag{1}$$

where $CE(\pi^{\text{exp}}, \pi_f)(S) = -\pi^{\text{exp}}(S) \odot \log \pi_f(S)$, $\odot$ denotes the usual dot-product, and $S$ is a random variable taking value $s \in \mathcal{S}$ with probability measure $\mu : \mathcal{S} \to [0, 1]$. Often $\mu(s)$ is chosen to equal the frequency with which an exploration policy (*e.g.*, random actions or $\pi^{\text{exp}}$) visits state $s$ in a randomly initialized episode. When it exists, we denote the policy minimizing Eq. (1) as $\pi_f^{\mu, \pi^{\text{exp}}}$. When $\mu$ and $\pi^{\text{exp}}$ are unambiguous, we write $\pi_f^{\text{IL}} = \pi_f^{\mu, \pi^{\text{exp}}}$.

What happens when there is a difference of privilege (or filtration functions) between the expert and the student? Intuitively, if the information that an expert uses to make a decision is unavailable to the student then the student has little hope of being able to mimic the expert's decisions. As we show in our next example, even when optimizing perfectly, depending on the choice of $f$ and $f^{\text{exp}}$, IL may result in $\pi_f^{\text{IL}}$ being uniformly random over a large collection of states. We call the phenomenon that $\pi_f^{\text{IL}} \neq \pi^{\text{exp}}$ the *imitation gap*.

**Example 2** (1D-Lighthouse). We illustrate the imitation gap using a gridworld spanning $[-N, \ldots, N]$. The two start states correspond to the goal at $-N$ or $N$, while the agent is always initialized at 0 (see Fig. 2a). Clearly, with full state information, $\pi^{\text{exp}}$ maps states to an 'always left' or 'always right' probability distribution, depending on whether the goal is on the left or right, respectively. Suppose now that the agent's visibility is constrained to a radius of $i$ (Fig. 2c shows $i = 1$), *i.e.*, an $f^i$-partial observation is accessible. An agent following an optimal policy with a visibility of radius $i$ will begin to move deterministically towards any corner, w.l.o.g. assume right. When the agent sees the rightmost edge (from position $N - i$), it will either continue to move right if the goal is visible or, if it's not, move left until it reaches the goal (at $-N$). Now we may ask: what is the best $f^i$-partial policy that can be learnt by imitating $\pi^{\text{exp}}$ (*i.e.*, what is $\pi_{f^i}^{\text{IL}}$)? *Tragically, the cross-entropy loss causes $\pi_{f^i}^{\text{IL}}$ to be uniform in a large number of states.* In particular, an agent following policy $\pi_{f^i}^{\text{IL}}$ will take left (and right) with probability 0.5, until it is within a distance of $i$ from one of the corners. Subsequently, it will head directly to the goal. See the policies highlighted in Figs. 2c, 2d. The intuition for this result is straightforward: until the agent observes one of the corners it cannot know if the goal is to the right or left and, conditional on its observations, each of these events is equally likely under $\mu$ (assumed uniform). Hence for half of these events the expert will instruct the agent to go right. For the other half the instruction is to go left. See App. A.1 for a rigorous treatment of this example. In Sec. 4 and Fig. 5, we train $f^i$-partial policies with $f^j$-optimal experts for a 2D variant of this example. We empirically verify that a student learns a better policy when imitating teachers whose filtration function is closest to their own.

The above example shows: when a student attempts to imitate an expert that is privileged with information not available to the student, the student learns a version of $\pi^{\text{exp}}$ in which this privileged information is marginalized out. We formalize this intuition in the following proposition.

**Proposition 1** (Policy Averaging). *In the setting of Section 3.1, suppose that $\Pi^{\text{feas.}}$ contains all $f$-partial policies. Then, for any $s \in \mathcal{S}$ with $o = f(s)$, we have that $\pi_f^{\text{IL}}(o) = \mathbb{E}_\mu[\pi^{\text{exp}}(S) \mid f(S) = o]$.*

Proofs are deferred to Appendix A.2.

The imitation gap provides theoretical justification for the common practical observation that an agent trained via IL can often be significantly improved by continuing to train the agent using pure RL (*e.g.*, PPO) (Lowe et al., 2020; Das et al., 2018b). Obviously training first with IL and then via pure RL techniques is ad hoc and potentially sub-optimal as discussed in Ex. 1 and empirically shown in Sec. 4. To alleviate this problem, the student should imitate the expert policy only in settings in which the expert policy can, in principle, be exactly reproduced by the student. Otherwise the student should learn via 'standard' RL methods. To this end, we introduce ADVISOR.

## 3.2 Adaptive Insubordination (ADVISOR) with Policy Gradients

To close the imitation gap, ADVISOR adaptively weights reward-based and imitation losses. Intuitively, it supervises a student to imitate an expert policy only in those states $s \in \mathcal{S}$ for which the imitation gap is small. For all other states, it trains the student using reward-based RL. To simplify notation, we denote the reward-based RL loss via $\mathbb{E}_\mu[L(\theta, S)]$ for some loss function $L$.[1] This loss formulation is general and spans all policy gradient methods, including A2C and PPO. The imitation loss is the standard cross-entropy loss $\mathbb{E}_\mu[CE(\pi^{\text{exp}}(S), \pi_f(S; \theta))]$. Concretely, ADVISOR loss is:

$$\mathcal{L}^{\text{ADV}}(\theta) = \mathbb{E}_\mu[w(S) \cdot CE(\pi^{\text{exp}}(S), \pi_f(S; \theta)) + (1 - w(S)) \cdot L(\theta, S)] \,. \tag{2}$$

Our goal is to find a *weight function* $w : \mathcal{S} \times \Theta \to [0, 1]$ where $w(s) \approx 1$ when the imitation gap is small and $w(s) \approx 0$ otherwise. For this we need an estimator of the distance between $\pi^{\text{exp}}$ and $\pi_f^{\text{IL}}$ at a state $s$ and a mapping from this distance to weights.

We now define $d^0(\pi, \pi_f)(s)$, a distance estimate between a policy $\pi$ and an $f$-partial policy $\pi_f$ at a state $s$. We can use any common non-negative distance (or divergence) $d$ between probability distributions on $\mathcal{A}$, *e.g.*, the KL-divergence (which we use in our experiments). While there are many possible strategies for using $d$ to estimate $d^0(\pi, \pi_f)(s)$, perhaps the simplest of these strategies is to define $d^0(\pi, \pi_f)(s) = d(\pi(s), \pi_f(s))$. Note that this quantity does not attempt to use any information about the fiber $f^{-1}(f(s))$ which may be useful in producing more holistic measures of distances.[2] Appendix A.3 considers how those distances can be used in lieu of $d^0$. Next, using the above, we need to estimate the quantity $d^0(\pi^{\text{exp}}, \pi_f^{\text{IL}})(s)$.

Unfortunately it is, in general, impossible to compute $d^0(\pi^{\text{exp}}, \pi_f^{\text{IL}})(s)$ exactly as it is intractable to compute the optimal minimizer $\pi_f^{\text{IL}}$. Instead we leverage an estimator of $\pi_f^{\text{IL}}$ which we term $\pi_f^{\text{aux}}$, and will define in the next section.

Given $\pi_f^{\text{aux}}$ we obtain the estimator $d^0(\pi^{\text{exp}}, \pi_f^{\text{aux}})$ of $d^0(\pi^{\text{exp}}, \pi_f^{\text{IL}})$. Additionally, we make use of the monotonically decreasing function $m_{\alpha,\beta} : \mathbb{R}_{\geq 0} \to [0, 1]$, where $\alpha, \beta \geq 0$. We define our weight function $w(s)$ for $s \in \mathcal{S}$ as:

$$w(s) = m_{\alpha,\beta}(d^0(\pi^{\text{exp}}, \pi_f^{\text{aux}})(s)) \quad \text{with} \tag{3}$$

$$m_{\alpha,\beta}(x) = e^{-\alpha x} \cdot 1_{[x \leq \beta]} \tag{4}$$

Together Eq. 2, 3, 4 define ADVISOR.

---

[1] For readability, we implicitly make three key simplifications. First, computing the expectation $\mathbb{E}_\mu[\ldots]$ is generally intractable, hence we cannot directly minimize losses such as $\mathbb{E}_\mu[L(\theta, S)]$. Instead, we approximate the expectation using rollouts from $\mu$ and optimize the empirical loss. Second, recent RL methods adjust the measure $\mu$ over states as optimization progresses while we assume it to be static for simplicity. Our final simplification regards the degree to which any loss can be, and is, optimized. In general, losses are often optimized by gradient descent and generally no guarantees are given that the global optimum can be found. Extending our presentation to encompass these issues is straightforward but notationally dense.

[2] Measures using such information include $\max_{s' \in f^{-1}(f(s))} d(\pi(s'), \pi_f(s))$ or a corresponding expectation instead of the maximization, *i.e.*, $\mathbb{E}_\mu[d(\pi(S), \pi_f(S)) \mid f(S) = o]$.

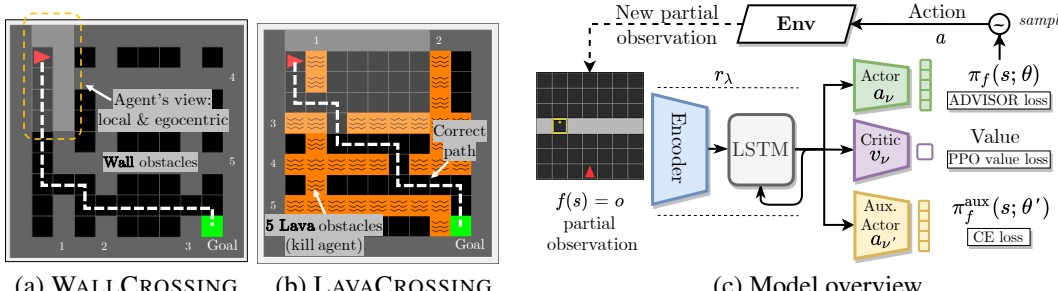

(a) WALLCROSSING    (b) LAVACROSSING    (c) Model overview

Figure 3: **MINIGRID base tasks and model overview.** (a) WC: Navigation with wall obstacles, with additional expert and environmental challenges. We test up-to $25 \times 25$ grids with 10 walls. (b) LC: Safe navigation, avoiding lethal lava rivers. We test up-to $15 \times 15$ grids with 10 lava rivers. (c) An auxiliary actor is added and trained only using IL. The 'main' actor policy is trained using the ADVISOR loss defined in Section 3.2, 3.3.

## 3.3 THE AUXILIARY POLICY $\pi^{\text{AUX}}$: ESTIMATING $\pi_f^{\text{IL}}$ IN PRACTICE

In this section we describe how we can, during training, obtain an *auxiliary policy* $\pi_f^{\text{aux}}$ which estimates $\pi_f^{\text{IL}}$. Given this auxiliary policy we estimate $d^0(\pi^{\text{exp}}, \pi_f^{\text{IL}})(s)$ using the plug-in estimator $d^0(\pi^{\text{exp}}, \pi_f^{\text{aux}})(s)$. While plug-in estimators are intuitive and simple to define, they need not be statistically efficient. In Appendix A.4 we consider possible strategies for improving the statistical efficiency of our plug-in estimator via prospective estimation.

In Fig. 3c we provide an overview of how we compute the estimator $\pi_f^{\text{aux}}$ via deep nets. As is common practice (Mnih et al., 2016; Heess et al., 2017; Jaderberg et al., 2017; Pathak et al., 2017; Mirowski et al., 2017; Chevalier-Boisvert et al., 2018a), the policy net $\pi_f(\cdot; \theta)$ is composed via $a_\nu \circ r_\lambda$ with $\theta = (\nu, \lambda)$, where $a_\nu$ is the *actor head* (possibly complemented in actor-critic models by a *critic head* $v_\nu$) and $r_\lambda$ is called the *representation network*. Generally, $a_\nu$ is lightweight, for instance a linear layer or a shallow MLP followed by a soft-max function. Instead, $r_\lambda$ is a deep and possibly recurrent neural net. We add another actor head $a_{\nu'}$ to our existing network which shares the underlying representation $r_\lambda$, *i.e.*, $\pi_f^{\text{aux}} = a_{\nu'} \circ r_\lambda$. As both actors share the representation $r_\lambda$ they benefit from any mutual learning. While our instantiation covers most use-cases, ADVISOR can be extended to estimating $\pi_f^{\text{IL}}$ via two separate networks, *i.e.*, $\theta' = (\nu', \lambda')$. In practice we train $\pi_f(\cdot; \theta)$ and $\pi_f^{\text{aux}}(\cdot; \theta)$ simultaneously using stochastic gradient descent, as summarized in Algorithm A.1.

## 4 EXPERIMENTS

We rigorously compare ADVISOR to IL methods, RL methods and their popularly-adopted (yet ad hoc) combinations. In particular, we evaluate 14 methods. We do this over ten tasks – realizations of Ex. 1 & Ex. 2, and eight navigational tasks of varying complexity within the fast, versatile MINIGRID environment (Chevalier-Boisvert et al., 2018a;b). Furthermore, for robustness, we train 50 hyperparameter variants for complex tasks. For all tasks, we find ADVISOR-based methods outperform or match performance of all baselines. In a final study, we train agents to complete the PointGoal Navigation task in the visually rich AIHABITAT environment. In this task we compare ADVISOR to prior work (using PPO) and pure-IL baselines.

### 4.1 TASKS

Succinct descriptions of our tasks follow. Experts always take globally optimal actions. We defer further details and description of experts to Appendix A.5.

**POISONEDDOORS (PD).** As defined in Ex. 1 in Sec. 1 with $N = 4$, $M = 10$, see Fig. 1. The agent's observation is an integer identifying the agent's state: (1) the agent must pick a door, (2) the agent has picked the first door and has either not started entering a code or has just previously entered a wrong input, or (3) the agent has picked the first door and its last action was a correct code input. The agent's action space includes 1 action for each door (used to pick among them) and three additional actions corresponding to input code values (only operational after choosing door 1).

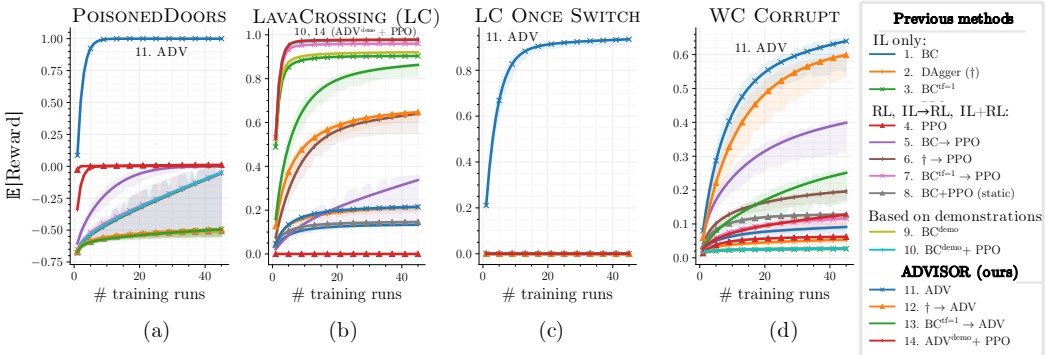

Figure 4: **Evaluation following (Dodge et al., 2019).** As described in Section 4.3, we plot expected validation reward of best-found model (y-axis) over an increasing budget of # training runs, each with random hyperparameter values (x-axis). Clearly, larger $\mathbb{E}[\text{Reward}]$ with fewer # training runs is better. We mark the best performing method(s) at the top of each plot.

**WALLCROSSING/LAVACROSSING (WC/LC).** As illustrated in Fig. 3a, 3b, an agent is tasked to navigate to the goal using local observations. In doing so, it must avoid walls or deadly (*i.e.*, episode-ending) rivers of lava. Evidently, imitating a shortest-path expert is easy, requiring no exploration beyond expert-visited states. Hence, we consider more challenging variants of WC and LC tasks. For all MINIGRID environments, agents observe a $7{\times}7{\times}3$ tensor corresponding to the area just in front of the agent. For our tasks only 3 actions are relevant (move ahead, rotate left, and rotate right). **SWITCH.** The agent is initialized in a WC or LC environment with the "lights turned off." The agent can use an additional *switch* action to get unaffected observations, whereas the shortest-path expert can navigate in the dark. Hence, the expert doesn't supervise taking the new action. For both WC and LC base tasks, we experiment with two switches: (1) lights stay on after using the *switch* action (ONCE), or (2) light turn on only for a single timestep (FAULTY). **CORRUPT.** To evaluate resilience of methods to a corrupted expert. In every episode the expert produces correct actions until it is within $N_C$ steps of the target, after which it outputs random actions. **2D-LIGHTHOUSE (2D-LH).** A harder, 2D variant of the gridworld task introduced in Ex. 2. Agents act by moving in a cardinal direction and observe some fixed radius about themselves.

## 4.2 BASELINES AND ADVISOR-BASED METHODS

Expert supervision comes in two forms: (a) as an expert policy, or (b) as a dataset of expert demonstrations. We study baselines and ADVISOR in both these forms. For (a), we include IL baselines with different levels of teacher-forcing (tf): tf=0, tf annealed from 1→0, and tf=1. This leads to Behaviour Cloning (BC), Data Aggregation (DAgger, †), and BC$^{\text{tf}=1}$, respectively. Also, we implement pure RL (PPO) which learns only on the sparse rewards. Furthermore, we implement popular sequential hybrids such as BC then PPO (BC→PPO), DAgger then PPO († → PPO), BC$^{\text{tf}=1}$ → PPO, and a parallel combination of BC + PPO(static). This is a static variant of our adaptive combination ADVISOR (ADV). We introduce hybrids such as DAgger then ADVISOR († → ADV), and BC$^{\text{tf}=1}$ → ADV. For (b), agents imitate expert demonstrations and hence get no supervision beyond the states in the demonstrations. This leads to BC$^{\text{demo}}$ and its combination with PPO (BC$^{\text{demo}}$ + PPO). We introduce a corresponding ADV$^{\text{demo}}$ + PPO, applying ADVISOR on expert demonstrations while training PPO on on-policy rollouts (see App. A.10 for details). Further details of all methods are in Appendix A.6. For fairness, the same model architecture is shared across all methods (recall Fig. 3c, Sec. 3.3). We defer implementation details to Appendix A.7.

## 4.3 EVALUATION

**Fair Hyperparameter Tuning.** Often unintentionally done, extensively tuning the hyperparameters (hps) of a proposed method and not those of the baselines can introduce unfair bias into evaluations. We avoid this by considering two strategies. For PD and all MINIGRID

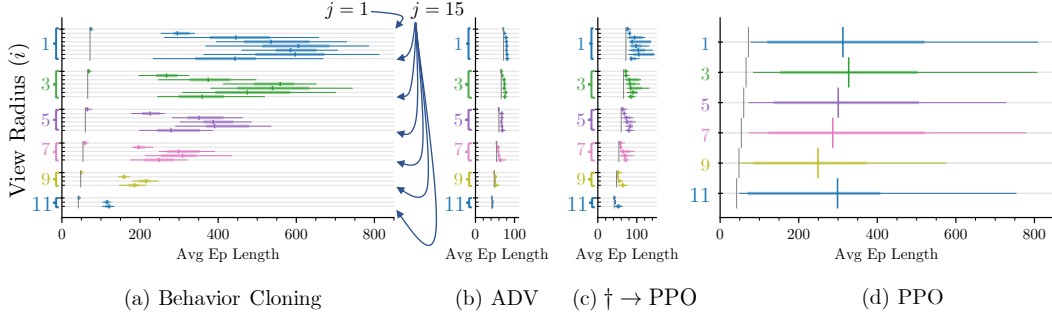

Figure 5: **"Less intelligent" teachers.** Learning $f^i$-partial policies using $f^j$-optimal experts 2D-LH.

tasks, we follow recent best practices (Dodge et al., 2019).[3] Namely, we tune each method by randomly sampling a fixed number of hps and reporting, for each baseline, an estimate of $\mathbb{E}$[Val. reward of best model when allowed a budget of $k$ random hps] for $1 \leq k \leq 45$. For this we must train 50 models per method, *i.e.*, 700 for each of these nine tasks. More details in Appendix A.8. For 2D-LH, we tune the hps of a competing method and use these hps for all other methods.

**Training.** For the eight MINIGRID tasks, we train each of the 50 training runs for 1 million steps. For 2D-LH/PD, models saturate much before $3 \cdot 10^5$ steps (details are in Appendix A.9).

**Metrics.** We record avg. rewards, episode lengths, and success rates. In the following, we report a subset of these recorded values. Additional plots can be found in Appendix A.11.

## 4.4 RESULTS

Before delving into task-specific analysis, we state three overall trends. First, for tasks where best-performing baselines require querying the expert policy during training, ADV significantly outperforms all methods. These are the more difficult tasks which need exploration, such as PD and SWITCH. Second, for tasks where the best-performing baselines are demonstrations-based, $\text{ADV}^{\text{demo}}$ + PPO improves over or matches previous methods. Third, a head-on comparison of BC + PPO(static) and ADV shows that our dynamic weighting approach is superior across all tasks.

**PD.** This environment was designed to be adversarial to standard imitation-learning approaches and so it is not surprising, see Fig. 4a, that models trained using standard IL techniques (DAgger, BC, $\text{BC}^{\text{tf}=1}$) perform poorly. Qualitatively, they attempt to open door 1 with a low probability and thus obtain an expected reward near $-2/3$. Baselines that learn from rewards, *e.g.*, PPO, can learn to avoid doors 2-4 but in practice cannot learn the combination to open the first door. This results in an average reward of 0. Notice that warm-starting with any form of IL is actively harmful: *e.g.*, it takes many hyperparameter evaluations before we consistently find a DAgger→PPO model that reproduces the performance of a plain PPO model. Finally, only our ADV method consistently produces high quality models (avg. reward approaching 1).

**LC.** The vanilla LC setting reveals that the imitation gap between the expert and the agent observations is nominal. Intuitively, the agent's egocentric partial observations are sufficient to learn to replicate the shortest-path expert actions: it need only follow alleys of safe ground leading to the narrow gaps in otherwise continuous obstacles. This is validated by the high performance of $\text{BC}^{\text{tf}=1}$ (and $\text{BC}^{\text{demo}}$) which learn only from expert trajectories and have no hope of bridging an imitation gap. Methods learning from demonstrations and RL (*e.g.*, $\text{ADV}^{\text{demo}}$+PPO and $\text{BC}^{\text{demo}}$ →PPO) perform only marginally better. Thus, when the imitation gap is small and BC from demonstrations is highly successful, we conclude that the gains from using ADVISOR-based methods may be marginal.

**LC SWITCH.** IL and warm-started methods receive no supervision to explore the *switch* action and thus learning in the dark resulting in poor policies, see Fig 4c. Also, early episode termination when agents encounter lava prevents PPO success due to sparse rewards. ADVISOR leverages it's RL loss to learn to 'switch' on lights after which it successfully imitates the expert.

**WC CORRUPT.** In Fig. 4d we investigate ADVISOR's ability to learn to ignore a corrupted expert. While this is not what ADVISOR was designed for, it is interesting to see that ADV-based methods are able to accomplish this task and do significantly better than the best performing competitor (BC→PPO). This suggests that ADVISOR is robust to expert failure.

---

[3] See also reproducibility checklist in EMNLP'20 CfP: https://2020.emnlp.org/call-for-papers

**2D-LH.** Here we vary the privilege of an expert and study learning from "less intelligent" teachers. Particularly, for each method, we train an $f^i$-partial policy using an $f^j$-optimal expert (except for PPO which uses no expert supervision) 25 times. Each policy is then evaluated on 200 random episodes and the average episode length (lower being better) is recorded. For all odd $i, j$ with $1 \leq i \leq 11$, $1 \leq j \leq 15$, and $j \geq i$ we show boxplots of the 25 training runs. Grey vertical lines show optimal average episode lengths for $f^i$-partial policies.

For BC we find training of an $f^i$-partial policy with an $f^j$-expert to result in a near optimal policy when $i = j$ but even small increases in $j$ result in dramatic decreases in performance. This emphasizes the imitation gap. Surprisingly, while performance tends to drop with increasing $j$, the largest $i, j$ gaps do not consistently correspond to the worst performing models. While this seems to differ from our results in Ex. 2, recall that there the policy $\mu$ was fixed while here it varies through training, resulting in complex learning dynamics.

Additionally, we find that: (i) PPO can perform well but has high variance across runs due to the problem of sparse rewards, and (ii) for this task, both ADVISOR and DAgger→PPO can ameliorate the impact of the imitation gap but ADVISOR consistently outperforms in all settings.

### 4.5 POINTGOAL NAVIGATION IN VISUALLY-RICH 3D ENVIRONMENTS

To highlight possible applications of ADVISOR to complex, visually rich, environments, we consider the PointGoal navigation (PointNav) task within AIHABITAT (Savva et al., 2019). We briefly describe the task, followed by the methods, and discuss results.

**Task**. In PointNav, a randomly spawned agent must navigate to a goal specified by a relative-displacement vector. The observation space is composed of rich egocentric RGB observations ($256 \times 256 \times 3$) with a limited field of view. The action space is {move_ahead, rotate_right, rotate_left, stop}. The task was formulated by Anderson et al. (2018) and implemented for the AIHABITAT simulator by Savva et al. (2019). Our reward structure, train/val/test splits, PointNav dataset, and implementation[4] follow Savva et al. (2019). To better understand sample-efficiency, we report the primary metric adopted to evaluate PointNav, *i.e.*, success weighted by path length (SPL), on the validation set at different points of training. We train on the standard Gibson set of 76 scenes, and report metrics as an average over the val. set consisting of 14 unseen scenes in AIHABITAT. We use a budget of 25 Mn frames, *i.e.*, ∼2 days of training on 4 NVIDIA TitanX GPUs, and 28 CPUs for each method.

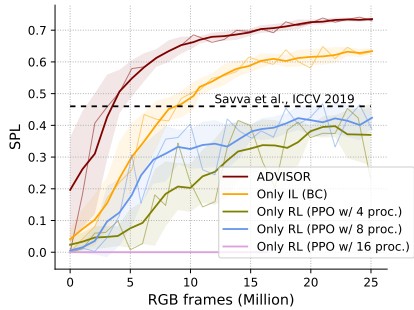

| Methods | SPL @5↑ | SPL @10↑ | SPL @25↑ |
|---|---|---|---|
| PPO (3 runs) | 0.307 | 0.404 | 0.468 |
| BC | 0.259 | 0.505 | 0.649 |
| ADVISOR | **0.579** | **0.685** | **0.741** |

Figure 6: PointNav in AIHABITAT.

**Methods**. Savva et al. (2019) train their RGB agents using PPO. We train pure IL (on-policy behavior cloning) using the optimal shortest-path action. The pure IL agent observes a filtered egocentric observation while the expert has access to the full env. state. We also train on-policy ADVISOR.

**Results.** Fig. 6 summarizes the results for the above methods. Despite being extensively tuned, we found that the training configuration released with habitat-lab is susceptible to small changes in the number of processes that collect data for PPO optimization. In contrast, for pure IL and ADVISOR baselines we don't tune hyper-parameters, *i.e.*, we use *exactly* the same hyper-parameters as the released configuration. We mark an SPL of $0.46$ via a dashed line, which was the result reported by Savva et al. (2019) for an RGB agent on the Gibson dataset. ADVISOR substantially outperforms other methods.

## 5 CONCLUSION

We introduce the *imitation gap* as one explanation for the empirical observation that imitating "more intelligent" teachers can lead to worse policies. While prior work has, implicitly, attempted to bridge this imitation gap, we introduce a principled adaptive weighting technique (ADVISOR), which we test on a suite of ten tasks. Due to the fast rendering speed of MINIGRID, PD and 2D-LH, we could undertake a study where we trained over 6 billion steps, to draw statistically significant inferences.

---

[4]https://github.com/facebookresearch/habitat-lab

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

APPENDIX: BRIDGING THE IMITATION GAP BY ADAPTIVE INSUBORDINATION

The appendix includes theoretical extensions of ideas presented in the main paper and details of empirical analysis. We structure the appendix into the following subsections:

A.9 Training implementation including maximum steps per episode, reward structure and computing infrastructure adopted for this work. We clearly summarize all structural and training hyperparameters for better reproducibility.

# A ADDITIONAL INFORMATION

## A.1 FORMAL TREATMENT OF EXAMPLE 2

Let $N \geq 1$ and consider a 1-dimensional grid-world with states $\mathcal{S} = \{-N, N\} \times \{0, \ldots, T\} \times \{-N, \ldots, N\}^T$. Here $g \in \{-N, N\}$ are possible goal positions, elements $t \in \{0, \ldots, T\}$ correspond to the episode's current timestep, and $(p_i)_{i=1}^T \in \{-N, \ldots, N\}^T$ correspond to possible agent trajectories of length $T$. Taking action $a \in \mathcal{A} = \{\text{left}, \text{right}\} = \{-1, 1\}$ in state $(g, t, (p_i)_{i=1}^T) \in \mathcal{S}$ results in the deterministic transition to state $(g, t+1, (p_1, \ldots, p_t, \text{clip}(p_t + a, -N, N), 0, \ldots, 0))$. An episode start state is chosen uniformly at random from the set $\{(\pm N, 0, (0, \ldots, 0))\}$ and the goal of the agent is to reach some state $(g, t, (p_i)_{i=1}^T)$ with $p_t = g$ in the fewest steps possible. We now consider a collection of filtration functions $f^i$, that allow the agent to see spaces up to $i$ steps left/right of its current position but otherwise has perfect memory of its actions. See Figs. 2c, 2d for examples of $f^1$- and $f^2$-partial observations. For $0 \leq i \leq N$ we define $f^i$ so that

$$f^i(g, t, (p_i)_{i=1}^T) = ((\ell_0, \ldots, \ell_t), (p_1 - p_0, \ldots, p_t - p_{t-1})) \quad \text{and} \tag{5}$$

$$\ell_j = (1_{[p_j + k = N]} - 1_{[p_j + k = -N]} \mid k \in \{-i, \ldots, i\}) \quad \text{for } 0 \leq j \leq t. \tag{6}$$

Here $\ell_j$ is a tuple of length $2 \cdot i + 1$ and corresponds to the agent's view at timestep $j$ while $p_{k+1} - p_k$ uniquely identifies the action taken by the agent at timestep $k$. Let $\pi^{\exp}$ be the optimal policy given full state information so that $\pi^{\exp}(g, t, (p_i)_{i=1}^T) = (1_{[g=-N]}, 1_{[g=N]})$ and let $\mu$ be a uniform distribution over states in $\mathcal{S}$. It is straightforward to show that an agent following policy $\pi_{f^i}^{\mathrm{IL}}$ will take random actions until it is within a distance of $i$ from one of the corners $\{-N, N\}$ after which it will head directly to the goal, see the policies highlighted in Figs. 2c, 2d. The intuition for this result is straightforward: until the agent observes one of the corners it cannot know if the goal is to the right or left and, conditional on its observations, each of these events is equally likely under $\mu$. Hence in half of these events the expert will instruct the agent to go right and in the other half to go left. The cross entropy loss will thus force $\pi_{f^i}^{\mathrm{IL}}$ to be uniform in all such states. Formally,

---

[5]We overload main paper's notation $d^0(\pi, \pi_f)(s)$ with $d_\pi^0(\pi_f)(s)$

---

**Algorithm A.1: On-policy ADVISOR algorithm overview.** Some details omitted for clarity.

**Input:** Trainable policies $(\pi_f, \pi_f^{\text{aux}})$, expert policy $\pi^{\text{exp}}$, rollout length $L$, environment $\mathcal{E}$.
**Output:** Trained policy

1 **begin**
2      Initialize the environment $\mathcal{E}$
3      $\theta \leftarrow$ randomly initialized parameters
4      **while** *Training completion criterion not met* **do**
5          Take $L$ steps in the environment using $\pi_f(\cdot; \theta)$ and record resulting rewards and observations (restarting $\mathcal{E}$ whenever the agent has reached a terminal state)
6          Evaluate $\pi_f^{\text{aux}}(\cdot; \theta)$ and $\pi^{\text{exp}}$ at each of the above steps
7          $L \leftarrow$ the empirical version of the loss from Eq. (2) computed using the above rollout
8          Compute $\nabla_\theta L$ using backpropagation
9          Update $\theta$ using $\nabla_\theta L$ via gradient descent
10      **return** $\pi_f(\cdot; \theta)$

---

we will have, for $s = (g, t, (p_i)_{i=1}^T)$, $\pi_{f^i}^{\text{IL}}(s) = \pi^{\text{exp}}(s)$ if and only if $\min_{0 \le q \le t}(p_q) - i \le -N$ or $\max_{0 \le q \le t}(p_q) + i \ge N$ and, for all other $s$, we have $\pi_{f^i}^{\text{IL}}(s) = (1/2, 1/2)$. In Sec. 4, see also Fig. 5, we train $f^i$-partial policies with $f^j$-optimal experts for a 2D variant of this example. ∎

### A.2 PROOF OF PROPOSITION 1

We wish to show that the minimizer of $\mathbb{E}_\mu[-\pi_{f^e}^{\text{exp}}(S) \odot \log \pi_f(S)]$ among all $f$-partial policies $\pi_f$ is the policy $\overline{\pi} = \mathbb{E}_\mu[\pi^{\text{exp}}(S) \mid f(S)]$. This is straightforward, by the law of iterated expectations and as $\pi_f(s) = \pi_f(f(s))$ by definition. We obtain

$$
\begin{aligned}
\mathbb{E}_\mu[-\pi_{f^e}^{\text{exp}}(S) \odot \log \pi_f(S)] &= -\mathbb{E}_\mu[E_\mu[\pi_{f^e}^{\text{exp}}(S) \odot \log \pi_f(S) \mid f(S)]] \\
&= -\mathbb{E}_\mu[E_\mu[\pi_{f^e}^{\text{exp}}(S) \odot \log \pi_f(f(S)) \mid f(S)]] \\
&= -\mathbb{E}_\mu[E_\mu[\pi_{f^e}^{\text{exp}}(S) \mid f(S)] \odot \log \pi_f(f(S))] \\
&= \mathbb{E}_\mu[-\overline{\pi}(f(S)) \odot \log \pi_f(f(S))] .
\end{aligned}
\tag{7}
$$

Now let $s \in \mathcal{S}$ and let $o = f(s)$. It is well known, by Gibbs' inequality, that $-\overline{\pi}(o) \odot \log \pi_f(o)$ is minimized (in $\pi_f(o)$) by letting $\pi_f(o) = \overline{\pi}(o)$ and this minimizer is feasible as we have assumed that $\Pi_f$ contains *all* $f$-partial policies. Hence it follows immediately that Eq. (7) is minimized by letting $\pi_f = \overline{\pi}$ which proves the claimed proposition.

### A.3 OTHER DISTANCE MEASURES

As discussed in Section 3.2, there are several different choices one may make when choosing a measure of distance between the expert policy $\pi^{\text{exp}}$ and an $f$-partial policy $\pi_f$ at a state $s \in \mathcal{S}$. The measure of distance we use in our experiments, $d_{\pi^{\text{exp}}}^0(\pi_f)(s) = d(\pi^{\text{exp}}(s), \pi_f(s))$, has the (potentially) undesirable property that $f(s) = f(s')$ does not imply that $d_{\pi^{\text{exp}}}^0(\pi_f)(s) = d_{\pi^{\text{exp}}}^0(\pi_f)(s')$. While an in-depth evaluation of the merits of different distance measures is beyond this current work, we suspect that a careful choice of such a distance measure may have a substantial impact on the speed of training. The following proposition lists a collection of possible distance measures with a conceptual illustration given in Fig. A.1.

**Proposition 2.** *Let $s \in \mathcal{S}$ and $o = f(s)$ and for any $0 < \beta < \infty$ define, for any policy $\pi$ and $f$-partial policy $\pi_f$,*

$$
d_{\mu,\pi}^\beta(\pi_f)(s) = E_\mu[(d_\pi^0(\pi_f)(S))^\beta \mid f(S) = f(s)]^{1/\beta},
\tag{8}
$$

*with $d_{\mu,\pi}^\infty(\pi_f)(s)$ equalling the essential supremum of $d_\pi^0(\pi_f)$ under the conditional distribution $P_\mu(\cdot \mid f(S) = f(s))$. As a special case note that*

$$
d_{\mu,\pi}^1(\pi_f)(s) = E_\mu[d_\pi^0(\pi_f)(S) \mid f(S) = f(s)].
$$

*Then, for all $\beta \geq 0$ and $s \in \mathcal{S}$ (almost surely $\mu$), we have that $\pi(s) \neq \pi_f(f(s))$ if and only if $d_\pi^\beta(\pi_f)(s) > 0$.*

*Proof.* This statement follows trivially from the definition of $\pi^{\mathrm{IL}}$ and the fact that $d(\pi, \pi') \geq 0$ with $d(\pi, \pi') = 0$ if and only if $\pi = \pi'$. $\qquad\square$

The above proposition shows that any $d^\beta$ can be used to consistently detect differences between $\pi^{\exp}$ and $\pi_f^{\mathrm{IL}}$, *i.e.*, it can be used to detect the imitation gap. Notice also that for any $\beta > 0$ we have that $d_{\mu,\pi^{\exp}}^\beta(\pi_f^{\mathrm{IL}})(s) = d_{\mu,\pi^{\exp}}^\beta(\pi_f^{\mathrm{IL}})(s')$ whenever $f(s) = f(s')$.

As an alternative to using $d^0$, we now describe how $d_{\mu,\pi^{\exp}}^1(\pi_f^{\mathrm{IL}})(s)$ can be estimated in practice during training. Let $\pi_f^{\mathrm{aux}}$ be an estimator of $\pi_f^{\mathrm{IL}}$ as usual. To estimate $d_{\mu,\pi^{\exp}}^1(\pi_f^{\mathrm{IL}})(s)$ we assume we have access to a function approximator $g_\psi : \mathcal{O}_f \to \mathbb{R}$ parameterized by $\psi \in \Psi$, *e.g.*, a neural network. Then we estimate $d_{\mu,\pi^{\exp}}^1(\pi_f^{\mathrm{IL}})(s)$ with $g_{\widehat{\psi}}$ where $\widehat{\psi}$ is taken to be the minimizer of the loss

$$\mathcal{L}_{\mu,\pi^{\exp},\pi_f^{\mathrm{aux}}}(\psi) = E_\mu\left[\left(d(\pi^{\exp}(S), \pi_f^{\mathrm{aux}}(f(S))) - g_\psi(f(S))\right)^2\right]. \tag{9}$$

The following proposition then shows that, assuming that $d_{\mu,\pi^{\exp}}^1(\pi_f^{\mathrm{aux}}) \in \{g_\psi \mid \psi \in \Psi\}$, $g_{\widehat{\psi}}$ will equal $d_{\mu,\pi^{\exp}}^1(\pi_f^{\mathrm{aux}})$ and thus $g_{\widehat{\psi}}$ may be interpreted as a plug-in estimator of $d_{\mu,\pi^{\exp}}^1(\pi_f^{\mathrm{IL}})$.

**Proposition 3.** *For any $\psi \in \Psi$,*

$$\mathcal{L}_{\mu,\pi^{exp},\pi_f^{aux}}(\psi) = E_\mu[(d_{\mu,\pi^{exp}}^1(\pi_f^{aux})(S) - g_\psi(f(S)))^2] + c,$$

*where $c = E_\mu[(d(\pi^{exp}(S), \pi^{aux}(f(S))) - d_{\mu,\pi^{exp},\widehat{}}^1(S))^2]$ is constant in $\psi$ and this implies that if $d_{\mu,\pi^{exp}}^1(\pi_f^{aux}) \in \{g_\psi \mid \psi \in \Psi\}$ then $g_{\widehat{\psi}} = d_{\mu,\pi^{exp}}^1(\pi_f^{aux})$.*

*Proof.* In the following we let $O_f = f(S)$. We now have that

$$E_\mu[\left(d(\pi^{\exp}(S), \pi_f^{\mathrm{aux}}(O_f)) - g_\psi(O_f)\right)^2]$$
$$= E_\mu[\left((d(\pi^{\exp}(S), \pi_f^{\mathrm{aux}}(O_f)) - d_{\mu,\pi^{\exp}}^1(\pi_f^{\mathrm{aux}})(S)) + (d_{\mu,\pi^{\exp}}^1(\pi_f^{\mathrm{aux}})(S) - g_\psi(O_f))\right)^2]$$
$$= E_\mu[(d(\pi^{\exp}(S), \pi_f^{\mathrm{aux}}(O_f)) - d_{\mu,\pi^{\exp}}^1(\pi_f^{\mathrm{aux}})(S))^2] + E_\mu[(d_{\mu,\pi^{\exp}}^1(\pi_f^{\mathrm{aux}})(S) - g_\psi(O_f)))^2]$$
$$+ 2 \cdot E_\mu[((d(\pi^{\exp}(S), \pi_f^{\mathrm{aux}}(O_f)) - d_{\mu,\pi^{\exp}}^1(\pi_f^{\mathrm{aux}})(S)) \cdot (d_{\mu,\pi^{\exp}}^1(\pi_f^{\mathrm{aux}})(S) - g_\psi(O_f)))]$$
$$= c + E_\mu[(d_{\mu,\pi^{\exp}}^1(\pi_f^{\mathrm{aux}})(S) - g_\psi(O_f)))^2]$$
$$+ 2 \cdot E_\mu[((d(\pi^{\exp}(S), \pi_f^{\mathrm{aux}}(O_f)) - d_{\mu,\pi^{\exp}}^1(\pi_f^{\mathrm{aux}})(S)) \cdot (d_{\mu,\pi^{\exp}}^1(\pi_f^{\mathrm{aux}})(S) - g_\psi(O_f)))].$$

Now as as $d_{\mu,\pi^{\exp}}^1(\pi_f^{\mathrm{aux}})(s) = d_{\mu,\pi^{\exp}}^1(\pi_f^{\mathrm{aux}})(s')$ for any $s, s'$ with $f(s) = f(s')$ we have that $d_{\mu,\pi^{\exp}}^1(\pi_f^{\mathrm{aux}})(S) - g_\psi(O_f)$ is constant conditional on $O_f$ and thus

$$E_\mu[(d(\pi^{\exp}(S), \pi_f^{\mathrm{aux}}(O_f)) - d_{\mu,\pi^{\exp}}^1(\pi_f^{\mathrm{aux}})(S)) \cdot (d_{\mu,\pi^{\exp}}^1(\pi_f^{\mathrm{aux}})(S) - g_\psi(O_f)) \mid O_f]$$
$$= E_\mu[(d(\pi^{\exp}(S), \pi_f^{\mathrm{aux}}(O_f)) - d_{\mu,\pi^{\exp}}^1(\pi_f^{\mathrm{aux}})(S) \mid O_f] \cdot E_\mu[d_{\mu,\pi^{\exp}}^1(\pi_f^{\mathrm{aux}})(S) - g_\psi(O_f) \mid O_f]$$
$$= E_\mu[d_{\mu,\pi^{\exp}}^1(\pi_f^{\mathrm{aux}})(S) - d_{\mu,\pi^{\exp}}^1(\pi_f^{\mathrm{aux}})(S) \mid O_f] \cdot E_\mu[d_{\mu,\pi^{\exp}}^1(\pi_f^{\mathrm{aux}})(S) - g_\psi(O_f) \mid O_f]$$
$$= 0.$$

Combining the above results and using the law of iterated expectations gives the desired result. $\quad\square$

## A.4 FUTURE DIRECTIONS IN IMPROVING DISTANCE ESTIMATORS

In this section we highlight possible directions towards improving the estimation of $d_{\pi^{\exp}}^0(\pi_f^{\mathrm{IL}})(s)$ for $s \in \mathcal{S}$. As a comprehensive study of these directions is beyond the scope of this work, our aim in this section is intuition over formality. We will focus on $d^0$ here but similar ideas can be extended to other distance measures, *e.g.*, those in Sec. A.3.

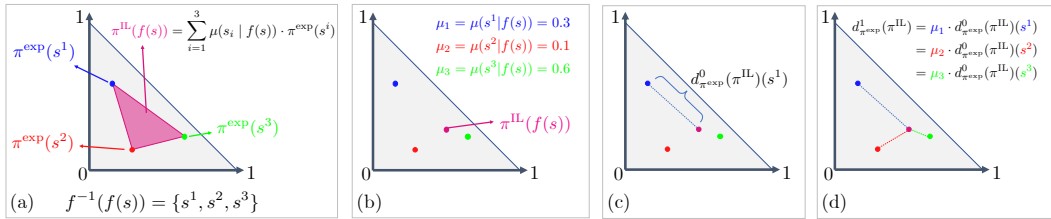

(a) $f^{-1}(f(s)) = \{s^1, s^2, s^3\}$   (b)   (c)   (d)

Figure A.1: **Concept Illustration.** Here we illustrate several of the concepts from our paper. Suppose our action space $\mathcal{A}$ contains three elements. Then for any $s \in \mathcal{S}$ and policy $\pi$, the value $\pi(s)$ can be represented as a single point in the 2-dimensional probability simplex $\{(x, y) \in \mathbb{R}^2 \mid x \geq 0, y \geq 0, x + y \leq 1\}$ shown as the grey area in (a). Suppose that the fiber $f^{-1}(f)$ contains the three unique states $s^1, s^2$, and $s^3$. In (a) we show the hypothetical values of $\pi^{\mathrm{exp}}$ when evaluated at these points. Proposition 1 says that $\pi^{\mathrm{IL}}(s)$ lies in the convex hull of $\{\pi^{\mathrm{exp}}(s^i)\}_{i=1}^3$ visualized as a magenta triangle in (a). Exactly where $\pi^{\mathrm{IL}}(s)$ lies depends on the probability measure $\mu$, in (b) we show how a particular instantiation of $\mu$ may result in a realization of $\pi^{\mathrm{IL}}(s)$ (not to scale). (c) shows how $d^1_{\pi^{\mathrm{exp}}}$ measures the distance between $\pi^{\mathrm{exp}}(s^1)$ and $\pi^{\mathrm{IL}}(s^1)$. Notice that it ignores $s^2$ and $s^3$. In (d), we illustrate how $d^0_{\pi^{\mathrm{exp}}}$ produces a "smoothed" measure of distance incorporating information about all $s^i$.

As discussed in the main paper, we estimate $d^0_{\pi^{\mathrm{exp}}}(\pi^{\mathrm{IL}}_f)(s)$ by first estimating $\pi^{\mathrm{IL}}_f$ with $\pi^{\mathrm{aux}}_f$ and then forming the "plug-in" estimator $d^0_{\pi^{\mathrm{exp}}}(\pi^{\mathrm{aux}}_f)(s)$. For brevity, we will write $d^0_{\pi^{\mathrm{exp}}}(\pi^{\mathrm{aux}}_f)(s)$ as $\widehat{d}$. While such plug-in estimators are easy to estimate and conceptually compelling, they need not be statistically efficient. Intuitively, the reason for this behavior is because we are spending too much effort in trying to create a high quality estimate $\pi^{\mathrm{aux}}_f$ of $\pi^{\mathrm{IL}}_f$ when we should be willing to sacrifice some of this quality in service of obtaining a better estimate of $d^0_{\pi^{\mathrm{exp}}}(\pi^{\mathrm{IL}}_f)(s)$. Very general work in this area has brought about the targeted maximum-likelihood estimation (TMLE) (van der Laan & Gruber, 2016) framework. Similar ideas may be fruitful in improving our estimator $\widehat{d}$.

Another weakness of $\widehat{d}$ discussed in the main paper is that is not prospective. In the main paper we assume, for readability, that we have trained the estimator $\pi^{\mathrm{aux}}_f$ before we train our main policy. In practice, we train $\pi^{\mathrm{aux}}_f$ alongside our main policy. Thus the quality of $\pi^{\mathrm{aux}}_f$ will improve throughout training. To clarify, suppose that, for $t \in [0, 1]$, $\pi^{\mathrm{aux}}_{f,t}$ is our estimate of $\pi^{\mathrm{IL}}_f$ after $(100 \cdot t)\%$ of training has completed. Now suppose that $(100 \cdot t)\%$ of training has completed and we wish to update our main policy using the ADVISOR loss given in Eq. (2). In our current approach we estimate $d^0_{\pi^{\mathrm{exp}}}(\pi^{\mathrm{IL}}_f)(s)$ using $d^0_{\pi^{\mathrm{exp}}}(\pi^{\mathrm{aux}}_{f,t})(s)$ when, ideally, we would prefer to use $d^0_{\pi^{\mathrm{exp}}}(\pi^{\mathrm{aux}}_{f,1})(s)$ from the end of training. Of course we will not know the value of $d^0_{\pi^{\mathrm{exp}}}(\pi^{\mathrm{aux}}_{f,1})(s)$ until the end of training but we can, in principle, use time-series methods to estimate it. To this end, let $q_\omega$ be a time-series model with parameters $\omega \in \Omega$ (e.g., $q_\omega$ might be a recurrent neural network) and suppose that we have stored the model checkpoints $(\pi^{\mathrm{aux}}_{f,i/K} \mid i/K \leq t)$. We can then train $q_\omega$ to perform forward prediction, for instance to minimize

$$\sum_{j=1}^{\lfloor t \cdot K \rfloor} \left( d^0_{\pi^{\mathrm{exp}}}(\pi^{\mathrm{aux}}_{f,j/K})(s) - q_\omega\left(s, (\pi^{\mathrm{aux}}_{f,i/K}(s))_{i=1}^{j-1}\right) \right)^2,$$

and then use this trained $q_\omega$ to predict the value of $d^0_{\pi^{\mathrm{exp}}}(\pi^{\mathrm{aux}}_{f,1})(s)$. The advantage of this prospective estimator $q_\omega$ is that it can detect that the auxiliary policy will eventually succeed in exactly imitating the expert in a given state and thus allow for supervising the main policy with the expert cross entropy loss earlier in training. The downside of such a method: it is significantly more complicated to implement and requires running inference using saved model checkpoints.

## A.5 ADDITIONAL TASK DETAILS

In Sec. 4.1, we introduced the different tasks where we compare ADVISOR with various other IL and RL methods. Here, we provide additional details for each of them along with information about observation space associated with each task. For training details for the tasks, please see Sec. A.9.

### A.5.1 POISONEDDOORS (PD)

This environment is a reproduction of our example from Sec. 1. An agent is presented with $N = 4$ doors $d_1, \ldots, d_4$. Door $d_1$ is locked, requiring a fixed $\{0, 1, 2\}^{10}$ code to open, but always results in a reward of 1 when opened. For some randomly chosen $j \in \{2, 3, 4\}$, opening door $d_j$ results in a reward of 2 and for $i \notin \{1, j\}$, opening door $d_i$ results in a reward of $-2$. The agent must first choose a door after which, if it has chosen door 1, it must enter the combination (receiving a reward of 0 if it enters the incorrect combination) and, otherwise, the agent immediately receives its reward. See Fig. 1.

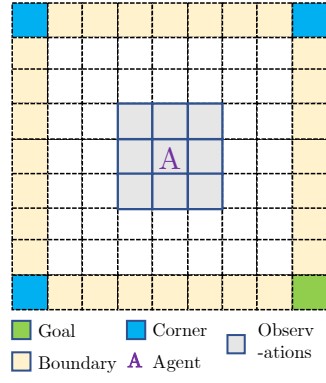

Figure A.2: 2D-LIGHTHOUSE

### A.5.2 2D-LIGHTHOUSE (2D-LH)

2D variant of the exemplar grid-world task introduced in Ex. 2, aimed to empirically verify our analysis of the imitation gap. A reward awaits at a randomly chosen corner of a square grid of size $2N + 1$ and the agent can only see the local region, a square of size $2i + 1$ about itself (an $f^i$-partial observation). Additionally, all $f^i$ allow the agent access to it's previous action. As explained in Ex. 2, we experiment with optimizing $f^i$-policies when given supervision from $f^j$-optimal experts (*i.e.*, experts that are optimal when restricted to $f^j$-partial observations). See Fig. A.2 for an illustration.

### A.5.3 WALLCROSSING (WC)

Initialized on the top-left corner the agent must navigate to the bottom-right goal location. There exists at least one path from start to end, navigating through obstacles. Refer to Fig. 3a where, for illustration, we show a simpler grid. Our environment is of size $25 \times 25$ with 10 `walls` ('S25, N10' as per the notation of (Chevalier-Boisvert et al., 2018b)), which are placed vertically or horizontally across the grid. The expert is a shortest path agent with access to the entire environment's connectivity graph and is implemented via the `networkx` python library.

### A.5.4 LAVACROSSING (LC)

Similar to WALLCROSSING in structure and expert, except that obstacles are `lava` instead of `walls`. Unlike `walls` (into which the agent can bump without consequence) here the episode terminates if the agent steps on `lava`. See Fig. 3b. This LC environment has size $25 \times 25$ with 10 `lava` rivers ('S25, N10').

### A.5.5 WC/LC SWITCH

In this task the agent faces a more challenging filtration function. In addition to navigational actions, agents for this task have a 'switch' action. Using this switch action, the agents can switch-on the lights of an otherwise darkened environment which is implemented as an observation tensor of all zeros. In WC, even in the dark, an agent can reach the target by taking random actions with non-negligible probability. Achieving this in LC is nearly impossible as random actions will, with high probability, result in stepping into `lava` and thereby immediately end the episode.

We experiment with two variants of this 'switch' – ONCE and FAULTY. In the ONCE SWITCH variant, once the the 'switch' action is taken, the lights remain on for the remainder of the episode. This is implemented as the unaffected observation tensor being available to the agent. In contrast, in the FAULTY SWITCH variant, taking the 'switch' action will only turn the lights on for a single timestep. This is implemented as observations being available for one timestep followed by zero tensors (unless the 'switch' action is executed again).

The expert for these tasks is the same as for WC and LC. Namely, the expert always takes actions along the shortest path from the agents current position to the goal and is unaffected by whether the light is on or off. For the expert-policy-based methods this translates to the learner agent getting perfect (navigational) supervision while struggling in the dark, with no cue for trying the switch action. For the expert-demonstrations-based methods this translates to the demonstrations being populated with blacked-out observations paired with perfect actions: such actions are, of course,

difficult to imitate. As FAULTY is more difficult than ONCE (and LC more difficult than WC) we set grid sizes to reduce the difference in difficulty between tasks. In particular, we choose to set WC ONCE SWITCH on a (S25, N10) grid and the LC ONCE SWITCH on a (S15, N7) grid. Moreover, WC FAULTY SWITCH is set with a (S15, N7) grid and LC FAULTY SWITCH with a (S9, N4) grid.

### A.5.6 WC/LC CORRUPT

In the SWITCH task, we study agents with observations affected by a challenging filtration function. In this task we experiment with corrupting the expert's actions. The expert policy flips over to a random policy when the expert is $N_C$ steps away from the goal. For the expert-policy-based method this translates to the expert outputting uniformly random actions once it is within $N_C$ steps from the target. For the expert-demonstrations-based methods this translates to the demonstrations consisting of some valid (observation, expert action) tuples, while the tuples close to the target have the expert action sampled from a uniform distribution over the action space. WC CORRUPT is a (S25, N10) grid with $N_C = 15$, while the LC CORRUPT is significantly harder, hence is a (S15, N7) grid with $N_C = 10$.

### A.5.7 OBSERVATION SPACE

Within our 2D-LH environment we wish to train our agent in the context of Proposition 1 so that the agent may learn any $f$-partial policy. As the 2D-LH environment is quite simple, we are able to uniquely encode the state observed by an agent using a $4^4 \cdot 5^2 = 6400$ dimensional $\{0, 1\}$-valued vector such that any $f$-partial policy can be represented as a linear function applied to this observation (followed by a soft-max).[6] Within the PD environment the agent's observed state is very simple: at every timestep the agent observes an element of $\{0, 1, 2, 3\}$ with 0 denoting that no door has yet been chosen, 1 denoting that the agent has chosen door $d_1$ but has not begun entering the code, 2 indicating that the agent has chosen door $d_1$ and has started entering the code, and 3 representing the final terminal state after a door has been opened or combination incorrectly entered. The MINIGRID environments (Chevalier-Boisvert et al., 2018b) enable agents with an egocentric "visual" observation which, in practice, is an integer tensor of shape $7 \times 7 \times 3$, where the channels contain integer labels corresponding to the cell's type, color, and state. Kindly see (Chevalier-Boisvert et al., 2018b;a) for details. For the above tasks, the cell types belong to the set of (`empty, lava, wall, goal`).

### A.6 ADDITIONAL BASELINE DETAILS

In Tab. A.1, we include details about the baselines considered in this work, including – purely IL $(1 - 3, 9)$, purely RL $(4)$, a sequential combination of them $(5 - 7)$, static combinations of them $(8, 10)$, and our dynamic combinations $(11 - 14)$. Moreover, we study methods which learn from both expert policy (expert action available for any state) and expert demonstrations (offline dataset of pre-collected trajectories). The hyperparameters (hps) we consider for optimization in our study have been chosen as those which, in preliminary experiments, had a substantial impact on model performance. This includes the learning rate (lr), portion of the training steps devoted to the first stage in methods with two stages (stage-split), and the temperature parameter in the weight function $(\alpha)$.[7] Implicitly, the random environment seed can also be seen as a hyperparameter. We sample hyperparameters uniformly at random. In particular, we sample lr from $[10^{-4}, 0.5)$ on a log-scale, stage-split from $[0.1, 0.9)$, and $\alpha$ from $\{5.0, 20.0\}$.

### A.7 ARCHITECTURE DETAILS

**2D-LH model.** As discussed in Sec. A.5.7, we have designed the observation given to our agent so that a simple linear layer followed by a soft-max function is sufficient to capture any $f$-partial policy. As such, our main and auxiliary actor models for this task are simply linear functions mapping

---

[6]As the softmax function prevents us from learning a truly deterministic policy we can only learn a policy arbitrarily close to such policies. In our setting, this distinction is irrelevant.

[7]See Sec. 3.2 for definition of the weight function for ADVISOR.

[8]While implemented with supervision from expert policy, due to the teacher forcing being set to 1.0, this method can never explore beyond states (and supervision) in expert demonstrations.

Table A.1: **Baseline details.** IL/RL: Nature of learning, Expert supervision: the type of expert supervision leveraged by each method, Hps. searched: hps. that were randomly searched over, fairly done with the same budget (see Sec. A.8 for details).

| # | Method | IL/RL | Expert supervision | Hps. searched |
|---|---|---|---|---|
| 1 | BC | IL | Policy | lr |
| 2 | † | IL | Policy | lr, stage-split |
| 3 | $BC^{tf=1}$ | IL | Policy[8] | lr |
| 4 | PPO | RL | Policy | lr |
| 5 | BC → PPO | IL→RL | Policy | lr, stage-split |
| 6 | † → PPO | IL→RL | Policy | lr, stage-split |
| 7 | $BC^{tf=1}$ → PPO | IL→RL | Policy | lr, stage-split |
| 8 | BC + PPO | IL+RL | Policy | lr |
| 9 | $BC^{demo}$ | IL | Demonstrations | lr |
| 10 | $BC^{demo}$ + PPO | IL+RL | Demonstrations | lr |
| 11 | ADV | IL+RL | Policy | lr, $\alpha$ |
| 12 | † → ADV | IL+RL | Policy | lr, $\alpha$, stage-split |
| 13 | $BC^{tf=1}$ → ADV | IL+RL | Policy | lr, $\alpha$, stage-split |
| 14 | $BC^{demo}$ + ADV | IL+RL | Demonstrations | lr, $\alpha$ |

the input 6400-dimensional observation to a 4-dimensional output vector followed by a soft-max non-linearity. The critic is computed similarly but with a 1-dimensional output and no non-linearity.

**PD model.** Our PD model has three sequential components. The first embedding layer maps a given observation, a value in $\{0, 1, 2, 3\}$, to an 128-dimensional embedding. This 128-dimensional vector is then fed into a 1-layer LSTM (with a 128-dimensional hidden state) to produce an 128-output representation $h$. We then compute our main actor policy by applying a $128 \times 7$ linear layer followed by a soft-max non-linearity. The auxiliary actor is produced similarly but with separate parameters in its linear layer. Finally the critic's value is generated by applying a $128 \times 1$ linear layer to $h$.

**MINIGRID model.** Here we detail each component of the model architecture illustrated in Fig. 3c. The encoder ('Enc.') converts observation tensors (integer tensor of shape $7 \times 7 \times 3$) to a corresponding embedding tensor via three embedding sets (of length 8) corresponding to type, color, and state of the object. The observation tensor, which represents the 'lights-out' condition, has a unique (*i.e.*, different from the ones listed by (Chevalier-Boisvert et al., 2018b)) type, color and state. This prevents any type, color or state from having more than one connotation. The output of the encoder is of size $7 \times 7 \times 24$. This tensor is flattened and fed into a (single-layered) LSTM with a 128-dimensional hidden space. The output of the LSTM is fed to the main actor, auxiliary actor, and the critic. All of these are single linear layers with output size of $|\mathcal{A}|$, $|\mathcal{A}|$ and 1, respectively (main and auxiliary actors are followed by soft-max non-linearities).

## A.8    FAIR HYPERPARAMETER TUNING

As discussed in the main paper, we consider two approaches for ensuring that comparisons to baselines are fair. In particular, we hope to avoid introducing misleading bias in our results by extensively tuning the hyperparameters (hps) of our ADVISOR methodology while leaving other methods relatively un-tuned.

**2D-LH: Tune by Tuning a Competing Method.** The goal of our experiments with the 2D-LH environment are, principally, to highlight that increasing the imitation gap can have a substantial detrimental impact on the quality of policies learned by training IL. Because of this, we wish to give IL the greatest opportunity to succeed and thus we are not, as in our other experiments, attempting to understand its expected performance when we must search for good hyperparameters. To this end, we perform the following procedure for every $i, j \in \{1, 3, 5 \dots, 15\}$ with $i < j$.

For every learning rate $\lambda \in \{100$ values evenly spaced in $[10^{-4}, 1]$ on a log-scale$\}$ we train a $f^i$-partial policy to imitate a $f^j$-optimal expert using BC. For each such trained policy, we roll out trajectories from the policy across 200 randomly sampled episodes (in the 2D-LH there is no

distinction between training, validation, and test episodes as there are only four unique initial world settings). For each rollout, we compute the average cross entropy between the learned policy and the expert's policy at every step. A "best" learning rate $\lambda^{i,j}$ is then chosen by selecting the learning rate resulting in the smallest cross entropy (after having smoothed the results with a locally-linear regression model (Wasserman, 2006)).

A final learning rate is then chosen as the average of the $\lambda^{i,j}$ and this learning rate is then used when training all methods to produce the plots in Fig. 5. As some baselines require additional hyperparameter choices, these other hyperparameters were chosen heuristically (post-hoc experiments suggest that results for the other methods are fairly robust to these other hyperparameters).

**All Other Tasks: Random Hyperparameter Evaluations.** As described in the main paper, we follow the best practices suggested by Dodge et al. (2019). In particular, for all tasks (except for 2D-LH) we train each of our baselines when sampling that method's hyperparameters, see Table A.1 and recall Sec. A.6, at random 50 times. Our plots, *e.g.*, Fig. 4, then report an estimate of the expected (validation set) performance of each of our methods when choosing the best performing model from a fixed number of random hyperparameter evaluations. Unlike (Dodge et al., 2019), we compute this estimate using a U-statistic (van der Vaart, 2000, Chapter 12) which is unbiased. Shaded regions encapsulate the 25-to-75th quantiles of the bootstrap distribution of this statistic.

### A.9    TRAINING IMPLEMENTATION

A summary of the training hyperparameters and their values is included in Tab. A.2. Kindly see (Schulman et al., 2017) for details on PPO and (Schulman et al., 2015b) for details on generalized advantage estimation (GAE).

**Max. steps per episode.** The maximum number of steps allowed in the 2D-LH task is 1000. Within the PD task, an agent can never take more than 11 steps in a single episode (1 action to select the door and then, at most, 10 more actions to input the combination if $d_1$ was selected) and thus we do not need to set a maximum number of allowed steps. The maximum steps allowed for an episode of WC/LC is set by (Chevalier-Boisvert et al., 2018b;a) to $4S^2$, where $S$ is the grid size. We share the same limits for the challenging variants – SWITCH and CORRUPT. Details of task variants, their grid size, and number of obstacles are included in Sec. A.5.

**Reward structure.** Within the 2D-LH task, the agent receives one of three possible rewards after every step: when the agent finds the goal it receives a reward of 0.99, if it otherwise has reached the maximum number of steps (1000) it receives a $-1$ reward, and otherwise, if neither of the prior cases hold, it obtains a reward of $-0.01$. See Sec. A.5.1 for a description of rewards for the PD task. For WC/LC, (Chevalier-Boisvert et al., 2018b;a) configure the environment to give a 0 reward unless the goal is reached. If the goal is reached, the reward is $1 - \frac{\text{episode length}}{\text{maximum steps}}$. We adopt the same reward structure for our SWITCH and CORRUPT variants as well.

**Computing infrastructure.** As mentioned in Sec. 4.3, for all tasks (except LH) we train 50 models (with randomly sampled hps) for each baseline. This amounts to 650 models per task or 5850 models in total. For each task, we utilize a `g4dn.12xlarge` instance on AWS consisting of 4 NVIDIA T4 GPUs and 48 CPUs. We run through a queue of 650 models using 48 processes. For tasks set in the MINIGRID environments, models each require $\approx 0.9$ GB GPU memory and all training completes in 18 to 24 hours. For the PD task, model memory footprints are smaller and training all 650 models is significantly faster ($< 8$ hours).

### A.10    THE ADV$^{\text{DEMO}}$ + PPO METHOD

As described in the main paper, the ADV$^{\text{demo}}$ + PPO method attempts to bring the benefits of our ADVISOR methodology to the setting where expert demonstrations are available but an expert policy (*i.e.*, an expert that can be evaluated at arbitrary states) is not. Attempting to compute the ADVISOR loss (recall Eq. (2)) on off-policy demonstrations is complicated however, as our RL loss assumes access to on-policy demonstrations. In theory, importance sampling methods, see, *e.g.*, (Mahmood et al., 2014), can be used to "reinterpret" expert demonstrations as though they were on-policy. But such methods are known to be somewhat unstable, non-trivial to implement, and may require information about the expert policy that we do not have access to. For these reasons, we choose to

Table A.2: Structural and training hyperparameters.

| Hyperparamter | Value |
|---|---|
| *Structural* | |
| Cell type embedding length | 8 |
| Cell color embedding length | 8 |
| Cell state embedding length | 8 |
| LSTM layers | 1 |
| LSTM hidden size | 128 |
| # Layers in critic | 1 |
| # Layers in actor | 1 |
| *PPO* | |
| Clip parameter ($\epsilon$) (Schulman et al., 2017) | 0.1 |
| Decay on $\epsilon$ | $\texttt{Linear}(1,0)$ |
| # Processes to sample steps | 20 |
| Rollout timesteps | 100 |
| Minibatch size | 1000 |
| Epochs | 4 |
| Value loss coefficient | 0.5 |
| Discount factor ($\gamma$) | 0.99 |
| GAE parameter ($\lambda$) | 1.0 |
| *Training* | |
| Optimizer | Adam (Kingma & Ba, 2017) |
| ($\beta_1, \beta_2$) for Adam | (0.9, 0.999) |
| Learning rate | $\texttt{searched}$ |
| Gradient clip norm | 0.5 |
| Training steps (WC/LC & variants) | $1 \cdot 10^6$ |
| Training steps (2D-LH & PD) | $3 \cdot 10^5$ |

use a simple solution: when computing the ADVISOR loss on expert demonstrations we ignore the RL loss. Thus $\text{ADV}^{\text{demo}} + \text{PPO}$ works by looping between two phases:

- Collect an (on-policy) rollout using the agent's policy, compute the PPO loss for this rollout and perform gradient descent on this loss to update the parameters.

- Sample a rollout from the expert demonstrations and, using this rollout, compute the demonstration-based ADVISOR loss

$$\mathcal{L}^{\text{ADV-demo}}(\theta) = \mathbb{E}_{\text{demos.}}[w(S) \cdot CE(\pi^{\text{exp}}(S), \pi_f(S; \theta))], \qquad (10)$$

and perform gradient descent on this loss to update the parameters.

## A.11 ADDITIONAL PLOTS

As mentioned in Sec. 4.3, we record three metrics for our tasks. Reward is the metric that best jointly captures success and effective path planning (see Sec. A.9 for reward structure). In Fig. A.3, we illustrate the key aspects of a figure showing expected maximum validation performance. In the main paper, we included some reward plots in Fig. 4. Specifically, Fig. A.4g, A.4e, and A.4d have already been included in the main paper (as Fig. 4c, 4b, and 4d). The remaining variants for WC/LC, FAULTY/ONCE SWITCH, and CORRUPT are presented in Fig. A.4.

Success rate shows a similar trend, following from the definition of rewards, *i.e.*, agents which reach the target more often, mostly end up with higher rewards. In Fig. A.5, we plot success rate for WC/LC, FAULTY/ONCE SWITCH, and CORRUPT tasks.

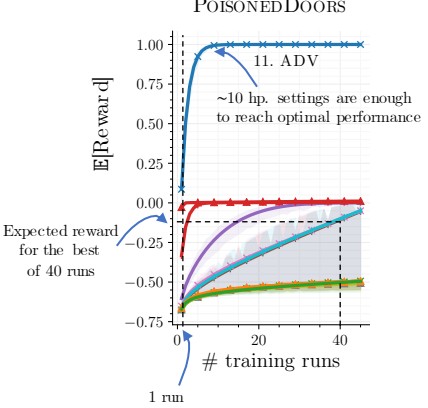

Figure A.3: Walk-through of Fig. 4.

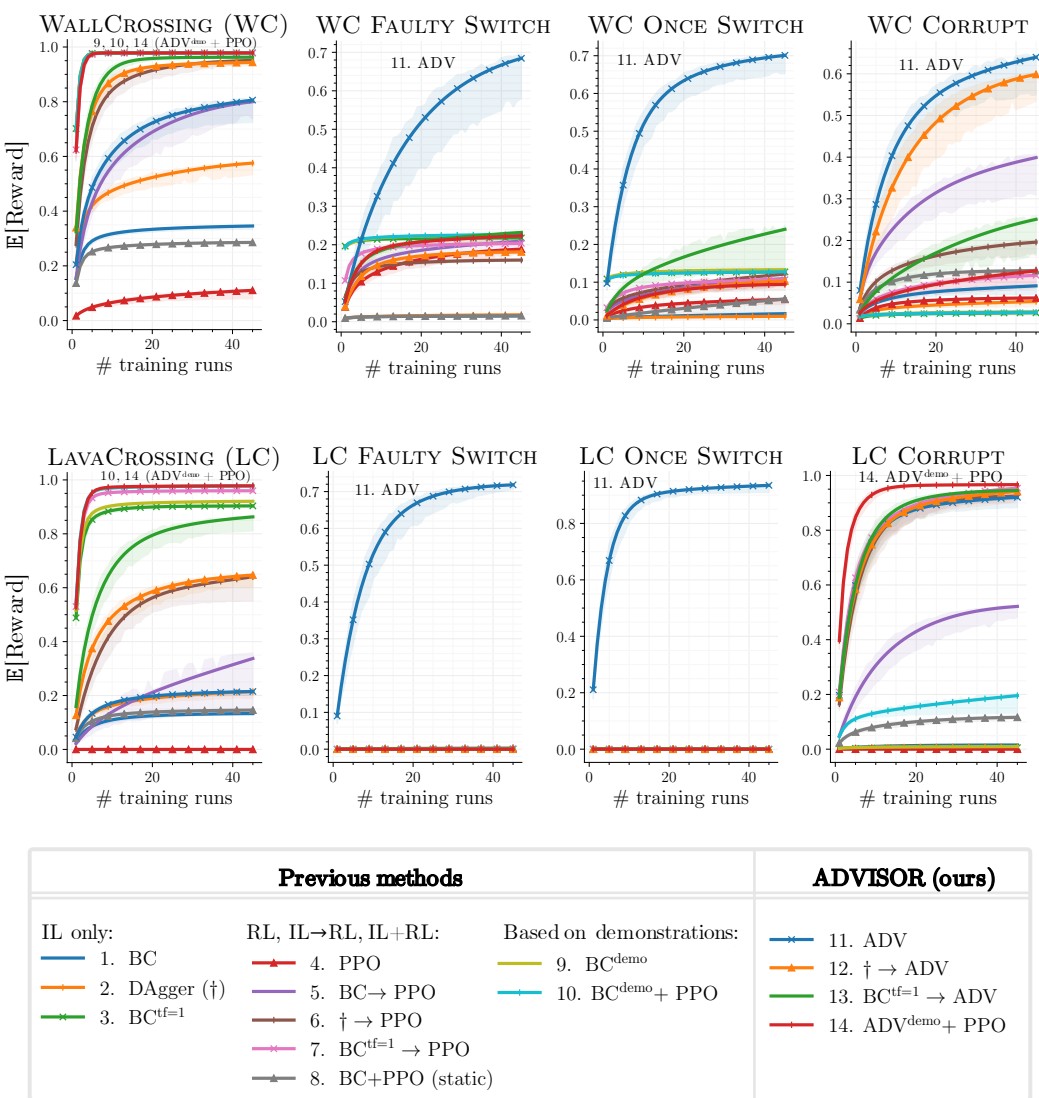

Figure A.4: $\mathbb{E}[\textbf{Reward}]$ **for baselines on MiniGrid tasks.** We include all variants of tasks considered in this work. Similar to Fig. 4, we plot estimates of the expected maximum validation set reward of all baselines (including our method), when allowing for increasingly many (random) hyperparameter evaluations (larger $\mathbb{E}[\text{Reward}]$ with fewer evals. is better).

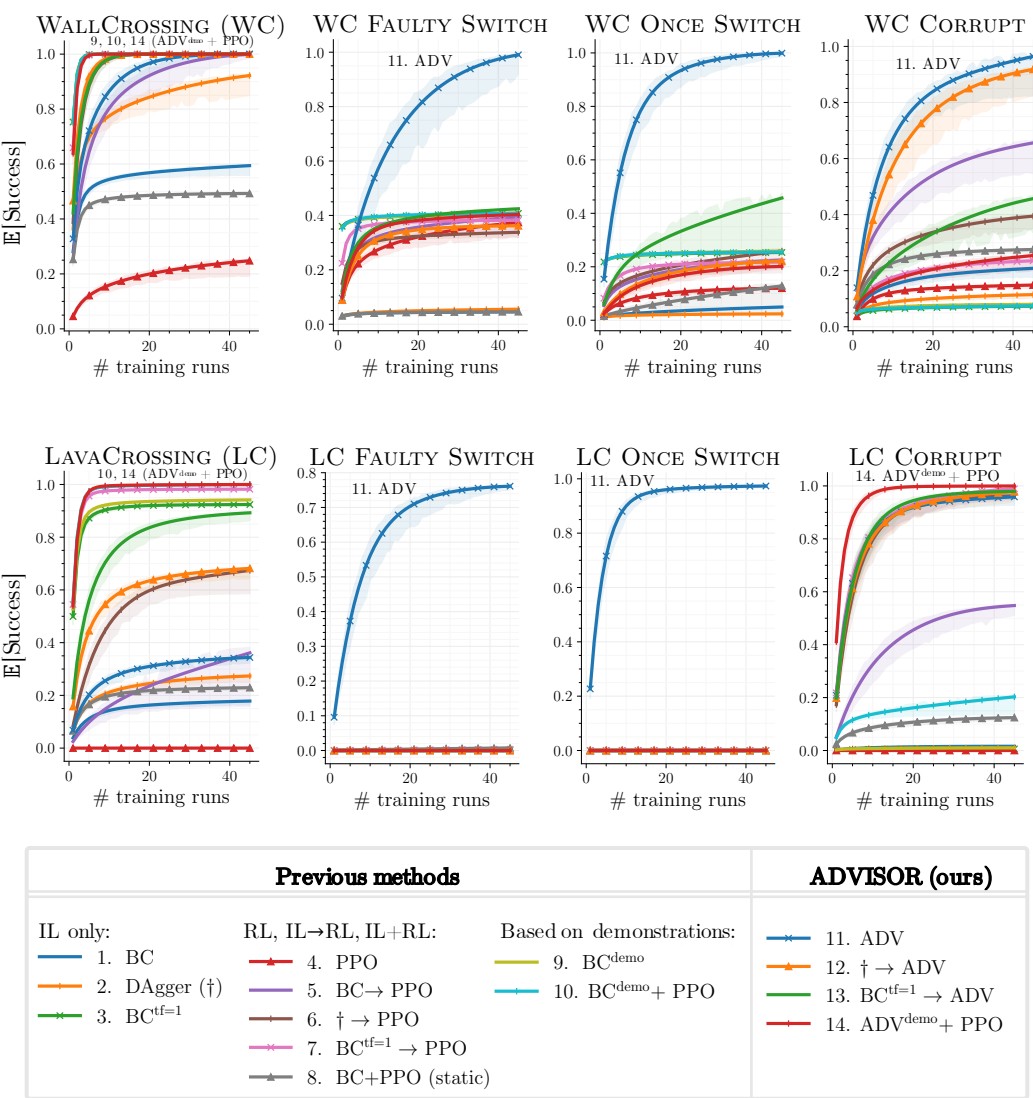

Figure A.5: $\mathbb{E}[$**Success Rate**$]$ **for baselines on MINIGRID tasks.** We include all variants of tasks considered in this work. This is the expected maximum validation set success rate of all baselines (including our method), when allowing for increasingly many (random) hyperparameter evaluations (larger $\mathbb{E}[$Success Rate$]$ with fewer evals. is better).

## A.12   IMPLEMENTATION DETAILS OF POINTNAV TASK

We follow the dataset splits released for the AIHabitat 2019 Challenge at CVPR 2019. The reward structure is identical to Savva et al. (2019). Concretely:

$$r_t = \begin{cases} r_{\text{goal}} + r_{\text{closer}} + \nu & \text{if agent 'stops' next to goal} \\ r_{\text{closer}} + \nu & \text{otherwise} \end{cases}. \tag{11}$$

$r_{\text{goal}}$ is the terminal reward of reaching the goal, $r_{\text{close}}$ is the decrease in the geodesic distance to the goal after the step taken at time $t$, and $\nu$ is the negative reward (time penalty) to encourage optimal paths. $r_{\text{goal}}$ is 10.0 and $\nu$ is $-0.01$, following the implementation of Savva et al. (2019) released as part of the `habitat-lab`.

To create Fig. 6, we evaluate checkpoints after every 1024k frames of experience. This is plotted as the thin line. The thick line and shading depicts the rolling mean (with a window size of 2) and corresponding standard deviation. We stopped training the PPO with 16 processes as it showed no signs of training. For the table, we state the best performance of checkpoints $\leq 5, 10, 25$ Mn frames of experience.

