# OpenReview forum: "Bridging the Imitation Gap by Adaptive Insubordination"
_ICLR.cc/2021/Conference — Reject_

### Official Review · AnonReviewer4 · 2020-10-25
**Review of: "Bridging the Imitation Gap by Adaptive Insubordination"**

**Rating:** 6
**Confidence:** 4

**Review:**

Short summary:

The paper presents an algorithm that allows an agent to leverage demonstrations from an expert with potential privileged  information compared to the agent. The main idea consists in measuring the distance or divergence between the imitation policy learned on the expert demonstrations and the expert policy and use this information to weight (convex combination) the importance of the imitation term and the reinforcement learning term in the combined loss function.

If there is indeed a discrepancy between the learned imitation policy and the expert policy, this could be linked to the fact that the expert and the agent do not share the same belief state.

 General comments:

The paper presents an important problem that could arise when naively applying imitation learning methods in partially observable MDPs where expert and apprentice do not share the same amount of information about the underlying state.  Basically, if the apprentice cannot distinguish states that are actually different for the expert, the imitation policy will be an average distribution of the expert policy over those states. This can be quite bad because the apprentice takes an average action instead of a precise one as it does not understand the underlying state as well as the expert. This is the imitation gap phenomenon.
However, it is also important to remind the reader that most imitation learning algorithms have been designed under the MDP hypothesis and therefore are not made to answer the imitation gap problems. It is only careless application of those algorithms in inappropriate settings such as POMDPs with asymmetric privileged information between expert and apprentice where this problem can arise. Therefore, I think the authors should mention some previous works that consisted in mitigating the problem of imitation gap such as in the field of conditional imitation learning:
- End-to-end Driving via Conditional Imitation Learning (https://arxiv.org/abs/1710.02410)
- Urban Driving with Conditional Imitation Learning (https://arxiv.org/abs/1912.00177)
where the goal is to add privileged information to the apprentice (in the form of navigational instructions) in order to have a better understanding of the underlying state. In addition the authors may also refer to works in representation learning for Imitation learning such as:
-  Learning Belief Representations for Imitation Learning in POMDPs (https://arxiv.org/abs/1906.09510)
that  try to provide better underlying state for the apprentice. Even if those works are not directly related to deal with imitation gap, they are partial solutions to it. Unlike the method proposed by the authors that consists in reducing the importance of demonstrations where there is an imitation gap, they directly try to improve the amount of information of the state provided to the apprentice.

Finally, concerning the method proposed by the authors, I do think that it is an interesting algorithm that weights demonstrations depending on the imitation gap. However this should be coupled with a method that try to enrich the state of the apprentice in order to not lose too much demonstrations. Using an LSTM is a first step towards that direction but more could be done (see papers provided in the previous section).

Detailed remarks:

I have several comments concerning specific sentences in the paper:
- "To overcome this imitation gap, prior work often uses stage-wise training": I do not agree with that at all. I think that previous work used stage-wise training mainly to improve upon the expert demonstrations. Indeed in some of the previous works,  expert and apprentice had directly access to the full state and therefore no imitation gap was possible. Still uncertainty on the expert policy could remain because demonstrations could come from different experts for instance.

-"As we show empirically, ADVISOR combines the benefits of IL and RL while avoiding the pitfalls of either method alone." I would also add that most of the IL methods were not designed to deal with the imitation gap as they make the MDP hypothesis.

-"These advances have been further improved upon through policy gradient methods", this has not really been shown or only on a very small set of control tasks. I think this should be reformulated because it sounds like policy gradient methods are better than deep-Q learning methods.

-"To the best of our knowledge, this hasn’t been studied before." weighting demonstrations depending on the imitation gap may not have been studied but methods that try to fill the gap have. It will be good that add this nuance.

- Concerning the imitation gap section, I would directly introduce the POMDP setting and explain simply the imitation gap there. Then I would go on formalising it. The example 2 description is too long and could be more concise and have more impact. I feel also the same with the algorithmic section that could be better organise.

Experiments:

The experiments are conducted in grid-worlds. They are interesting to show the intuition behind the algorithm. However, it will be easier to grant acceptance if the experiments were on visually rich 3D environnements or in robotics.

Rating: I think I could increase my rating if the authors take into account my remarks. I also believe that it is important to not only discard demonstrations where the information gap is important but to try to close the gap by better representation learning or memories. In addition a careful use and collection of expert demonstrations could also mitigate this problem.

---

> ### Author Response · Authors · 2020-11-24
> **Response to Reviewer #4**
>
> Thank you for your careful read of our work and your willingness to consider improving the rating conditional on your suggestions. We respond to your comments below.
>
> **Referencing IL works that more closely address the imitation gap**
>
> We have clarified in our related work section that, unlike prior work, we do not make an MDP assumption that would necessarily preclude the imitation gap. Thank you for these additional citations, they are clearly very relevant and are now included in our related work.
>
> **ADVISOR should be coupled with enriching the apprentice's state to retain as many demonstrations as possible**
>
> We absolutely agree and have added additional content to our related work to clarify this point.
>
> **Prior work uses stage-wise training mainly to improve upon expert demonstrations rather than overcome the imitation gap**
>
> This is a fair point and we have reworded the relevant sentence. We would argue however that there are real-world cases, e.g., visual navigation, where experts can be created in code to be perfect/deterministic and for which there would be a clear imitation gap. An expert with the map of a house would suggest actions following the shortest path with no exploration or grounding to the limited first-person observations of the apprentice.
>
> **Add remark that most IL methods are not designed to deal with the imitation gap**
>
> We have clarified this in Sec. 1.
>
> **Suggestion that policy gradient methods are better than deep-Q learning methods**
>
> We have rewritten this sentence in Sec. 2, first paragraph.
>
> **Adding nuance to our discussion of other methods and how they address the imitation gap.**
>
> We have added this nuance in Sec. 2.
>
> **Organizational/clarity improvement suggestions for the imitation gap section, example 2, and the algorithm section.**
>
> We have a streamlined example 2 to minimize technical details and increase its impact. We agree that Section 3 would benefit from some reorganization. As such changes would result in the appearance of relatively large changes to an entire section (and ICLR allows reviewers/ACs to ignore such changes) we felt it safer to keep the structure during the rebuttal period. We are happy to incorporate further structural/clarity changes in the camera-ready if requested.
>
> **Easier to grant acceptance if the experiments were on visually rich 3D environments or in robotics**
>
> As suggested, we have now applied our methods to another task in visually rich 3D environments. Particularly, we choose one of the widely-studied tasks in the computer vision community -- PointGoal Navigation within the AIHabitat simulator (Savva et al., 2019). Note that the Gibson dataset and AIHabitat simulator used for these experiments render based on real-world scans.  In the newly added Sec. 4.3, we report ADVISOR trains faster and better than pure RL and pure IL baselines. ADVISOR reached higher performance (~2x of IL or RL) early in training (5 Mn steps). At the end of training (25 Mn steps), ADVISOR performs 60% better than PPO (Savva et al. 2019) and 14% better than pure IL. We trained all our methods for 2 days on 4 NVIDIA TitanX GPUs and 28 CPUs. Other implementation details for the task are included in Sec. A.12.
>
> Since this suggestion mirrors the suggestion from reviewer #3, kindly refer to our response to reviewer #3, for additional discussion.
>
> Savva, Manolis, Abhishek Kadian, Oleksandr Maksymets, Yili Zhao, Erik Wijmans, Bhavana Jain, Julian Straub et al. "Habitat: A platform for embodied ai research." In Proceedings of the IEEE International Conference on Computer Vision, pp. 9339-9347. 2019.

---

### Official Review · AnonReviewer1 · 2020-10-27
**Review for paper "Bridging the Imitation Gap by Adaptive Insubordination"**

**Rating:** 6
**Confidence:** 4

**Review:**

[Summary]

Paper aims at attacking the "imitation gap" in the canonical LfD problem, where the expert and learner have different observation spaces, leading to poor performance if merely adopting imitation learning without self-exploration. A novel learning-based
weight proposing function is introduced to mitigate the issue by dynamically weighing the objective of imitation learning (specifically, behavior cloning) and reinforcement learning. The weight is computed based on whether the expert policy disagrees with the best imitation policy on a certain state. Experiments on several artificial tasks demonstrate the effectiveness of the proposed method over pure RL, pure IL, and some canonical RL+IL methodologies.

[Strength]

+) Personally, I like the research problem presented here. Compared to its counterparts on imitation learning with imperfect demonstrations, the authors did an excellent job of making their "imperfectness" concrete and also meaningful to potential real-world applications. I won't say it is universally applied but with the intuitive examples, the incentives and the resulting solutions are way more comprehensive than those focusing on injecting noise or perturbation into the demonstrations.

+) The paper is overall clear and well-written. I can't find any major technical issue, while some of the notation could be confusing; Nonetheless, I feel I can get the main idea after reading the main text plus the appendix.

+) The experiments sufficient enough to validate the proposed method. A combination of an artificial but contrived task (PD) and a challenging mini-grid task seems sufficient enough for me.

[Weakness]

The main concern I have with this submission lies in some missing factors and their potentials to the rationale of the main approach presented here.

-) First thing first, it seems that eq.2 (supposed to be the key loss of the proposed method) implicitly assumes that pi_f is ideal in terms of an imitated policy, so as to the only factor that accounts for the gap between pi_f and pi^exp is the misalignment of observation space. However, I find this assumption could be unrealistic as there are several factors that could affect the imitation performances as well, to name a few: the number of demonstrations (major), different imitation learning algorithms (major), and even the training strategy (epochs, etc, but minor though). The authors did not investigate these factors in their experiments and therefore, it is still unclear whether the loss in eq.2 makes sense as few demonstrations and underfitting will also possibly lead to the disagreement between these two policies. The authors are encouraged to conduct some ablation study with diff. number of demonstrations and also imitation learning methods other than BC to verify how these factors could affect their approach.

-) In the appendix, the authors emphasize that they "spending too much effort in trying to create a high-quality estimate of pi^aux to pi_f", while if my understanding is correct, they achieve this by co-train pi_aux with an imitation-only object along with the training of other models. I'm wondering why not just train pi_aux first then query it in the training of the rest? Is there any specific reason to sacrifice the estimation accuracy with co-training? Will the proposed approach enjoy better performance with this pre-train strategy?

-) According to eq.2, the computation of the weight requires a query on state s to both pi_f and pi^exp, while the authors introduce a demonstration-only variant in their experiments. As only the action is marginalized out, it could be tricky to do such computation merely with samples of (s, a). The authors are expected to provide more details on their implementation.

[Suggestions&Questions]

(1) Add an ablation study on how the number of demonstrations and imitation learning algorithm could affect the performances.

(2) Add a baseline that only pre-train pi^aux rather than co-train with other models.

(3) How can eq.2 work with a demonstration-only expert? Please clarify.

(4) Some citations need to be added:

Reinforcement learning from demonstration through shaping, in IJCAI, 2015

Direct policy iteration with demonstrations, in IJCAI, 2015

Policy shaping with human teachers, in IJCAI, 2015

Truncated horizon policy search: Combining reinforcement learning and imitation learning, in ICLR, 2018

Reinforcement Learning from Imperfect Demonstrations under Soft Expert Guidance, in AAAI, 2020

[Post-rebuttal]

I have read through all the other reviews and the rebuttal. Would like to thank the authors for their efforts in improving this submission. I  do believe some of my concerns have been addressed while the main issue on the lack of some necessary evaluations (e.g. num. of demonstrations) remains. Thus I may not be able to escalate my justification to even higher.

---

> ### Author Response · Authors · 2020-11-24
> **Response to Reviewer #1**
>
> Thank you for your thoughtful comments on our work and support of our submission. We provide point-by-point responses to your suggestions and questions below.
>
> **1. Ablation study to test the robustness of Eq. (3).**
>
> We agree that if the auxiliary policy is poorly optimized then it will provide little benefit during training. Indeed if the auxiliary policy remains completely unoptimized (so that it is uniformly random) then for sane choices of alpha, beta, Eq. (3) will effectively revert to a pure RL loss.
>
> Fundamentally, we feel that with this question you hope to see an answer to the question of if ADVISOR is applicable in more "realistic" settings where "realistic" here is with respect to the availability of expert feedback/optimization problems. With the compute resources we had available during this rebuttal period we focused our attention on this question in the context of more complex/visually-rich environments (as requested by two other reviewers). Please our response to reviewer #3. While this experiment does not directly address your question as posed, it hopefully provides some assurance that our method does work in practical settings and will be of value to the community. We plan to consider these other ablations in future revisions.
>
> **2. Baseline where $\pi^{\text{aux}}$ is pretrained.**
>
> Please note that all of our pipelined methods involving ADVISOR (e.g., BC$\to$ADV, DAgger$\to$ADV, etc.) do pretrain $\pi^{\text{aux}}$ (along with the main policy) and then continue training $\pi^{\text{aux}}$ during the remainder of training. In the following, we assume that when you say we should pretrain $\pi^{\text{aux}}$ you mean that we should do this pretraining and then freeze the weights associated with $\pi^{\text{aux}}$ when training the main policy.
>
> There are a few subtleties when pretraining $\pi^{\text{aux}}$, namely:
>
> * Which exploration policy do we use (i.e., do we use teacher forcing, follow $\pi^{\text{aux}}$, take random actions, etc.)?
> * Will a $\pi^{\text{aux}}$ trained with this exploration policy work when applied to the actions followed by the main policy?
>
> It is straightforward to come up with examples where a pretrained $\pi^{\text{aux}}$ will fail terribly when confronted with settings that it did not see during training. For instance, in our poisoned doors task (Example 1), if we train the auxiliary policy using teacher forcing, then it will learn to place 0 probability mass on the first door and equal probability mass on all other doors. As using teacher forcing means that we always take the expert's actions (one of doors 2, 3, or 4, when N=4), the $\pi^{\text{aux}}$ policy will never obtain supervision for what code to enter if it were to take door 1. Altogether, this means that when we train the main policy using this pretrained and frozen $\pi^{\text{aux}}$, we will be completely unable to learn which code to enter when taking the first door.
>
> Due to the above issues, time constraints, and as we do not expect pre-trained+frozen auxiliary models to outperform those that are pretrained and then co-trained (included in our baselines) we save these ablations for future work.
>
> **3. How do we compute Eq. (2) for the demonstration-only variant.**
>
> Thank you for pointing out that this was not described in the text. This is now described in Appendix A.10.
>
> In short: we use a two-stage process that loops between sampling expert demonstration and on-policy demonstration. When using expert demonstrations we apply our reweighing of IL losses as usual and simply ignore the RL loss. When using the on-policy demonstrations we apply the PPO loss as usual and do not attempt to reweigh it.
>
> **4. Related work**
>
> Thank you for these additional citations, we have now included them in our related work.

---

### Official Review · AnonReviewer3 · 2020-10-28
**Introducing a critical issue in imitation learning with a simple solution**

**Rating:** 6
**Confidence:** 3

**Review:**

### Summary
This paper identifies a problem in imitation learning when an expert has access to privileged information that is not available to the learner. When a decision has to be made based on the privileged information, the learner tends to choose average or uniformly random actions of the expert due to the lack of important information, which is called the "imitation gap". Therefore, in such cases, learning from the expert can actually harm the training of the learner.
This paper proposes to tackle this imitation gap by following the expert only when the learner can reproduce the expert's behaviors only with partial observations. Otherwise, the learner ignores the expert's guidance and relies on reward-based RL. The proposed method effectively balances between learning from the expert and learning from reward with the divergence between action distributions between expert and an auxiliary policy, which is trained sorely from the expert. The exhaustive experiments prove that the proposed method outperforms baselines in environments with a large imitation gap.

### Strengths
- This paper identifies the "imitation gap" that could be critical for many imitation learning applications with intuitive examples.
- The proposed method is simple but effective in resolving the imitation gap by learning from the expert only when the learner does not require access to privileged information.
- The comparative experiments are thoroughly conducted and well presented.
- The proposed method has many potential applications, such as sim2real transfer, where a policy trained in simulation has access to state information, and thus the imitation gap can be problematic.

### Weaknesses
- As the proposed method involves multiple learnable components, providing the algorithm could clarify how the method works. Currently, it is not clear to understand how the auxiliary policy is trained together with the imitation learning policy.
- The tasks tested in the paper are limited to simple 2D examples, which are tailored to the proposed method. Including more general and complex tasks could improve the work. It would be great to see how it can be generally applicable to continuous action space, e.g., ant navigation or manipulation tasks with partial observations or cluttered environments.
- ADV fails to achieve good performance (stuck around 0.2) on the LC task. In theory, ADV should learn as much as BC-demo+PPO and ADV-demo+PPO can. More explanation would help to understand the low performance of ADV compared to other methods.
- The observation and action spaces can be briefly described in the main paper. For example, the action space for PoisonedDoor is not presented in the paper.

### Questions and additional feedback
- Cite a relevant paper "Zhu et al. Reinforcement and Imitation Learning for Diverse Visuomotor Control, RSS 2018", which combines GAIL and RL, and demonstrates its effectiveness on diverse manipulation tasks.
- What is the reference for "teacher-forcing"?
- Attach the appendix at the end of the main paper.

---

> ### Author Response · Authors · 2020-11-24
> **Response to Reviewer #3**
>
> Thank you for your detailed review and support for our paper. We respond to each of your points below.
>
> **Clarifying the algorithm**
>
> We agree that an explicit algorithm would be helpful for clarity. This is now included as Algorithm A.1 in our updated PDF. After adding the below experiment with visually rich environments we, unfortunately, did not have sufficient space to include this in the main document. We will attempt to reorganize to create space in the camera-ready.
>
> **More complex tasks**
>
> We agree the observation space of our tasks is simple. However, our tasks are intricate, as evidenced by no method reaching optimal performance over many tasks. As mentioned at the start of Sec. 4 and in the conclusion, these simple simulators are fast and versatile (customizable and procedurally-generated), and permits to conduct a study where we trained over 6 billion steps of experience. As suggested, we applied our approach to another complex task with continuous state space. We choose one of the widely-studied tasks in the computer vision community -- PointGoal Navigation within the AIHabitat simulator (Savva et al., 2019). The task satisfies the following recommended requirements:
>
> * The agent operates in a *continuous state space* (up to machine precision).
> * AIHabitat provides a *complex observation space*, namely egocentric RGB images within photorealistic 3D scans of real-world houses.
> * The egocentric observations have a limited field of view leading to severe *partial observability*.
> * Scenes in AIHabitat are cluttered containing *several obstacles* and forward-only agents do poorly.
>
> Note, however, that the action space is still discrete. In the newly added Sec. 4.3, we report ADVISOR trains faster and better than pure RL and pure IL baselines. ADVISOR reached higher performance (~2x of IL or RL) early in training (5 Mn steps). At the end of training (25 Mn steps), ADVISOR performs 60% better than PPO (Savva et al., 2019) and 14% better than pure IL. We trained all our methods for 2 days on 4 NVIDIA TitanX GPUs and 28 CPUs. Other implementation details for the task are included in Sec. A.12.
>
> **Why do we get poor performance on the LC task?**
>
> Note that in these LC tasks we see that:
>
> * All purely on-policy methods, i.e., those with no teacher forcing or expert demonstrations, do poorly (≤0.4 success).
> * Methods using DAgger can do better than purely on-policy methods but also do far worse than the best-performing methods (≤0.7 success).
> * All best performing methods include an expert demonstration or an extended period of imitation learning with a teacher forcing of 1 (e.g., a teacher forcing of 1 is essentially equivalent to having a large dataset of expert demonstrations).
>
> The above is consistent with the hypothesis that the LC task:
>
> 1. Has little to no imitation gap (as pure IL is able to solve this task).
> 2. Is adversarial to on-policy methods.
>
> This second point is quite intuitive: recall that the LC map is quite large and perilous (a single wrong step will send the agent into lava and end the episode). This means that an on-policy agent will almost never accidentally obtain a reward (or even come near the goal) and thus it is extremely difficult to learn a sensible policy. While it is true that ADV does not perform spectacularly on this task, it performs as well or better than most of its on-policy counterparts (PPO does not train at all, and BC without teacher forcing is significantly worse). The only on-policy method that appears to outperform it is BC$\to$PPO. But note that the error bars for this method are quite large suggesting that it may simply have had a single lucky good run among the many hyperparameter evaluations.
>
> As you note, our variant of ADV which includes off-policy demonstrations (ADV-demo+PPO) does as well or better than all other methods.
>
> **Clarifying observation and action spaces**
>
> Our updated PDF now includes concise descriptions of these spaces, see the "Tasks" section.
>
> **Related work**
>
> Thank you for this reference, we have now included it in our discussion of related work.
>
> **Additional feedback**
>
> To the best of our knowledge (Williams and Zipser, 1989) introduced the term _teacher forcing_. It refers to the use of the ground-truth outputs to predict later outputs, instead of the predictions by the model. This has been commonly used in trajectory prediction and language applications (Leblond et al., 2018, Li et al. 2019). The annealed variant has also been referred to as _scheduled sampling_ in the literature (Bengio et al., 2015).
>
> We have attached the appendix to the main document.

---

> > ### Author Response · Authors · 2020-11-24
> > **References**
> >
> > Due to the character limit, we include the references below:
> >
> > Savva, Manolis, Abhishek Kadian, Oleksandr Maksymets, Yili Zhao, Erik Wijmans, Bhavana Jain, Julian Straub et al. "Habitat: A platform for embodied ai research." In Proceedings of the IEEE International Conference on Computer Vision, pp. 9339-9347. 2019.
> > Williams, Ronald J., and David Zipser. "A learning algorithm for continually running fully recurrent neural networks." Neural computation 1.2 (1989): 270-280.
> >
> > Leblond, Rémi, Jean-Baptiste Alayrac, Anton Osokin, and Simon Lacoste-Julien. "SEARNN: Training RNNs with global-local losses." In International Conference on Learning Representations. 2018.
> >
> > Li, Xiujun, Chunyuan Li, Qiaolin Xia, Yonatan Bisk, Asli Celikyilmaz, Jianfeng Gao, Noah A. Smith, and Yejin Choi. "Robust Navigation with Language Pretraining and Stochastic Sampling." In Proceedings of the 2019 Conference on Empirical Methods in Natural Language Processing and the 9th International Joint Conference on Natural Language Processing (EMNLP-IJCNLP), pp. 1494-1499. 2019.
> >
> > Bengio, Samy, Oriol Vinyals, Navdeep Jaitly, and Noam Shazeer. "Scheduled sampling for sequence prediction with recurrent neural networks." In Advances in Neural Information Processing Systems, pp. 1171-1179. 2015.

---

### Official Review · AnonReviewer2 · 2020-10-29
**Bridging the Imitation Gap by Adaptive Insubordination**

**Rating:** 5
**Confidence:** 3

**Review:**

Pros:
1. this paper studies an interesting problem, "imitation gap" in imitation learning.
2. the paper is easy to follow. Especially, the example part is easy to understand.
3. the intuition for this idea is well-explained.


Cons:
1. from my perspective, the basic idea is dynamically using imitation learning and reinforcement learning for agent learning by a weighting function. It is a straightforward idea but lacks some novelty.

2. the main contribution of this paper is proposing an advisor and integrating it with the imitation and reinforcement learning process. However, only averaging loss to leverage imitation and exploration is a comparatively little contribution.

3. in experiment PD, adding advisor from the beginning seems not fair enough, since at the beginning the reward is much higher than baselines. Maybe the experiments need a comparison before the advisor is added and after the advisor added.

In summary, in this paper, the motivation is clear and easy to understand, and the problem is worthy to study. But the contribution of this paper is a little limited. A better solution is needed.

---

> ### Comment · Program_Chairs · 2020-11-14
> **Misplaced review**
>
> This review seems to be misplaced, as it is not relevant to this submission. Please ignore this review.
>
> The area chair has left a message for the reviewer, and if they do not replace this with the correct review, we will remove this review from this forum.
>
> Thank you.

---

> ### Author Response · Authors · 2020-11-24
> **Response to Reviewer #2**
>
> Thank you for your time and feedback. We are happy to see that you feel we are studying an interesting problem and that our paper is well-written. We feel that there may have been some understandable but **critical misunderstandings of our work** which we clarify below.
>
> **Does our work have limited novelty?**
>
> We worry that there may have been some misunderstanding of our central method. It is true that we are reweighing losses coming from imitation and reinforcement learning but, critically, this reweighing is done _dynamically_ throughout the training. Namely, at every step, we use our understanding of the imitation gap (which, to the best of our knowledge, we are the first to explicitly study) to choose which loss to use. Moreover, we outperform the “averaging loss” or _static_ baseline, see `8. BC+PPO (static)` in Fig. 4, 8, and 9 -- emphasizing that _dynamic_ weighting is effective. We feel that this unifies several interesting ideas and offers a novel perspective on the dangers of standard imitation learning approaches. Our appendix discusses some ideas to potentially improve our method (see appendix A.3), but, as our experiments show, our current approach already works quite well and is easy to implement.
>
> **Unfairness as ADVISOR's reward is much higher than baselines at the start of training**
>
> _Thankfully, this worry is simply a misunderstanding of our graphs._
>
> To the best of our understanding, you are making reference to the fact that, in Fig. 4(a), the ADV method has a much larger reward than other methods at the beginning of the plot. This seems to suggest that the ADV method was given an unfair advantage (e.g., maybe a random weight initialization just happened to be very good). Critically, the **x-axis in this plot does not correspond to training steps but actually the number of full, independent, training runs (each trained fully for 1 Mn steps)**. Consequently, the first point in the plot refers to the performance of methods after randomly choosing one set of hyperparameters. Please refer to the discussion in Sec 4.3 and the newly added Fig. A.3 in the Appendix. Briefly summarizing: to ensure our experiments are as fair as possible, we are following the methodology from (Dodge et al., 2019) which evaluates a method by asking "how well would a method do if I ran it $k$ times with randomly selected hyperparameters?" The x-axis here is the number of hyperparameter settings used to train a method and the y-axis is the expected reward of the best model for those $k$ evaluations. Each line in the plot is an outcome of $50$ models, each trained for 1Mn steps. This is an incredibly fair way of evaluating methods and goes far beyond what is normally done when comparing RL algorithms.
>
> In light of the above, the fact that our method starts higher than the highest value of other methods in Fig. 4(a) emphasizes the strength of our method: this means that a randomly chosen set of hyperparameter values with our method works better than all other methods even when letting them try 45 different sets of hyperparameters.
>
> Dodge, Jesse, Suchin Gururangan, Dallas Card, Roy Schwartz, and Noah A. Smith. "Show Your Work: Improved Reporting of Experimental Results." In Proceedings of the 2019 Conference on Empirical Methods in Natural Language Processing and the 9th International Joint Conference on Natural Language Processing (EMNLP-IJCNLP), pp. 2185-2194. 2019.

---

### Decision · Program_Chairs · 2021-01-07
**Final Decision**

**Decision:**

Reject

**Comment:**

Reviewers acknowledged that the problem addressed in this paper is interesting and is not solved by the existing literature. They appreciated that the setup was well defined and the paper was clearly written. Yet they kept several concerns after the rebuttal. Especially, they expected the comparison to be done with algorithms using both demonstrations and rewards and the current empirical evaluation was not judged as fair. Also, the simple baseline consisting of adding an LSTM to BC to integrate past observations has not been considered either. This baseline is still missing to assess the quality of the proposed method.